# PlioMIP2 simulations using the MIROC4m climate model

Wing-Le Chan[1] and Ayako Abe-Ouchi[1,2]

[1]Atmosphere and Ocean Research Institute, The University of Tokyo, Kashiwa, 277-8564, Japan
[2]National Institute for Polar Research, Tachikawa, 190-8518, Japan

*Correspondence to*: Wing-Le Chan (wlchan@aori.u-tokyo.ac.jp)

**Abstract.** The second phase of the Pliocene Model Intercomparison Project (PlioMIP2) has attracted many climate modelling groups in its continuing efforts to better understand the climate of the mid-Piacenzian warm period (mPWP) when atmospheric $CO_2$ was last closest to present day levels. Like the first phase, PlioMIP1, it is an internationally coordinated initiative that allows for a systematic comparison of various models in a similar manner to the Paleoclimate Modelling Intercomparison
Project (PMIP). Model intercomparison and model-data comparison now focus specifically on the interglacial at marine isotope stage KM5c (3.205Ma) and experimental design is not only based on new boundary conditions but includes various sensitivity experiments. In this study, we present results from long-term model integrations using the MIROC4m atmosphere-ocean coupled general circulation model, developed at the institutes CCSR/NIES/FRCGC in Japan. The core experiment, with $CO_2$ levels set to 400ppm, shows a warming of 3.1°C compared to the Pre-Industrial, with two-thirds of the warming being
attributed to the increase in $CO_2$. Although this level of warming is less than that in the equivalent PlioMIP1 experiment, there is slightly better agreement with proxy sea surface temperature (SST) data at PRISM3 locations, especially in the northern North Atlantic where there were large model-data discrepancies in PlioMIP1. Similar spatial changes in precipitation and sea ice are seen and the Arctic remains ice-free in the summer in the core experiments of both phases. Comparisons with both the proxy SST data and proxy surface air temperature data from paleobotanical sites indicate a weaker polar amplification in model
results. Unlike PlioMIP1, the Atlantic Meridional Overturning Circulation (AMOC) is now stronger than that of the Pre-Industrial, even though increasing $CO_2$ tends to weaken it. This stronger AMOC is a consequence of a closed Bering Strait in the PlioMIP2 paleogeography. Also, when present day boundary conditions are replaced by those of the Pliocene, the dependency of the AMOC strength on $CO_2$ is significantly weakened. Sensitivity tests show that lower values of $CO_2$ give a global SST which is overall more consistent with the PRISM3 SST field presented in PlioMIP1, while SST at many of the
PRISM4 sites are still too high to be reconciled with any of the model results. On the other hand, tropical Pacific SST in the core experiment agrees well with more recent proxy data which suggested that PRISM3 SST there was overestimated. Future availability of climate reconstructions from proxy data will continue to help evaluate model results. Inclusion of dynamical vegetation and the effects of all possible extreme orbital configurations outside KM5c should be considered in future experiments using MIROC4m for the mPWP.

# 1 Introduction

The mid-Pliocene was the most recent period in the earth's history to have experienced sustained levels of atmospheric $CO_2$ similar to those of present day. Global temperatures are also estimated to have been 2-3°C higher than those of present day (Chandler et al., 2008). Various proxy evidence, including pollen assemblages (Brigham-Grette et al., 2013) and tetraether lipids (Crampton-Flood, 2017), have been used to reconstruct larger temperature changes at northern high latitudes. As a clear understanding of climate change in the near future becomes ever more important, scientists have also looked to warm climates of the past to validate model predictions and to quantify the forcings responsible for large climatic shifts. The mid-Pliocene has often been thought of as a good analogue for near-future climates, with formal assessments based on quantitative comparisons between past warm periods and Representative Concentration Pathway emission scenarios (Burke et al., 2018). The first studies of the mid-Pliocene using climate models were those of Chandler et al. (1994) whose results using the GISS AGCM showed a decrease in the equator-to-pole temperature gradient and of Sloan et al. (1996) who found a global surface temperature increase of 3.6°C with the NCAR GENESIS AGCM. These were eventually followed by studies using coupled atmosphere-ocean models, such as HadCM3 (Haywood and Valdes, 2004; Lunt et al., 2008) and NCAR CCMS3 (Jochum et al., 2009). Some of these studies investigated the specific effects of changing seaways and mountain uplifts which occurred during that time. Simultaneously, there have been on-going efforts with reconstructions of the mid-Pliocene climate using a multitude of marine and terrestrial proxy data. PRISM (Pliocene Research Interpretation and Synoptic Mapping) reconstructions of sea surface temperature (SST) and vegetation were first introduced in Dowsett et al. (1994) and are still on-going. Pollen assemblages from marine sediments have been used in Panitz et al. (2015) to reconstruct fluctuations of cool temperate and boreal conditions in northern Norway between 3.6 and 3.14 Ma. Plant macrofossils preserved in sediments across the Canadian Arctic Archipelago (Fletcher et al., 2017) have shown warmer and wetter conditions up to the mid-Pliocene in that region.

The Pliocene Model Intercomparison Project (PlioMIP) was initiated at a time when climate models were increasingly used to simulate past climates to assess their performances by way of model intercomparison or model-data comparison (e.g. Braconnot et al., 2007). The first phase (Haywood et al., 2010, 2011), henceforth named PlioMIP1, focused on the mid-Piacenzian warm period (mPWP) within the mid-Pliocene, defined by PRISM as the interval between 3.264 and 3.025Ma. Adhering to the experimental designs specified in PlioMIP1 and using PRISM3D boundary conditions, models simulated an annual mean surface temperature increase of 1.8 to 3.6°C from Pre-Industrial values, in addition to the aforementioned decrease in the equator-to-pole temperature gradient Haywood et al. (2013). Despite this polar amplification, models were unable to replicate the scale of warming in the northern North Atlantic as suggested by PRISM3 proxy data (Dowsett et al., 2013). Subsequent model intercomparison studies yielded findings on the intensification of the East Asian monsoon during the mPWP (Zhang, R. et al., 2013), the different responses of the Atlantic Meridional Overturning Circulation (Zhang, Z. et al., 2013) and the importance of albedo feedbacks to high latitude warming (Hill et al., 2014), amongst other topics.

Lessons learnt and new insights gained from PlioMIP1 were applied to the second phase of the project, PlioMIP2 (Haywood et al., 2016), and an attempt was made to address shortcomings from the design in the first phase, namely, uncertainties in the boundary conditions, data and model physics, which could contribute to model-data discord. New reconstructions for paleogeography, land surface elevation and ice sheet distribution as part of PRISM4 were used as boundary conditions for PlioMIP2 experiments (Dowsett et al., 2016), including closure of the Bering Strait and a smaller Greenland ice sheet restricted to the eastern part of the island. As opposed to a time slab or interval, the reconstructions now focus on a particular time slice, the interglacial peak MIS KM5c (3.205Ma), which has an orbital forcing close to that of present day. Additionally, PlioMIP2 forms part of the fourth phase of the Paleoclimate Modelling Intercomparison Project (PMIP4) and the sixth phase of the Coupled Model Intercomparison Project (CMIP6) (Kageyama et al, 2018).

PlioMIP2 also laid out plans for non-core sensitivity experiments to investigate the effects of individual boundary conditions and to account for the uncertainty in greenhouse gas levels. Estimates of $CO_2$ for this period from different marine proxy records vary widely, from 250 to 450ppm (Fedorov et al., 2013), including uncertainties in individual sources. Values in the upper half of that range are inferred from alkenone-derived $\delta^{13}C$ and foraminiferal $\delta^{11}B$ (Seki et al., 2010; Martínez-Botí et al., 2015), stomatal properties of leaves (Kürschner et al., 1996) and marine plankton $\delta^{13}C$ (Raymo et al., 1996). Estimates from the lower half are derived from B/Ca ratios in foraminifera (Tripati et al., 2009) and other foraminiferal $\delta^{11}B$ (Bartoli et al., 2011). The core experiment in PlioMIP2 has a setting of 400ppm for $CO_2$, which accounts for greenhouse gas forcing from all sources, and is close to the PlioMIP1 value.

The main aim of this study is to present results of both core and some non-core PlioMIP2 experiments using MIROC4m. With these results, we investigate the differences in the climate state when switching from PlioMIP1 boundary conditions to PlioMIP2. Using results from non-core experiments, we examine the individual effects of increasing $CO_2$ and of introducing PlioMIP2 boundary conditions. We also compare proxy data with our model results, including those from sensitivity experiments using a range of $CO_2$ levels.

## 2 Model description

In the present study, the experiments have been carried out with the coupled atmosphere-ocean general circulation model, MIROC4m (The Model for Interdisciplinary Research on Climate), a mid-resolution model developed jointly by the institutes CCSR, FRCGC and NIES in Japan (K-1 model developers, 2004). This model has previously been used to study a variety of climate states, for example, future RCP4.5 and RCP8.5 scenarios (Bakker et al., 2016), the mid-Holocene (Ohgaito et al, 2013), the Last Glacial Maximum (Yanase and Abe-Ouchi, 2007) and the mPWP (Chan et al., 2011). The same model with a higher resolution was included in the Fifth Assessment Report of the Intergovernmental Panel on Climate Change (IPCC, 2013). Note that this model is not a contributing member to PMIP4/CMIP6, and so results in this study are confined to PlioMIP2.

The model consists of an atmosphere-land-river component and a sea ice-ocean component, with the air-sea exchange of momentum, heat and water occurring at the air-sea ice interface. At ice-free grid cells, exchange still occurs via the sea ice subcomponent but the flux to the ocean remains unaffected by the sea ice. Below is a brief description and readers should refer to K-1 model developers (2004) and the references contained within for further details.

The atmospheric component is identical to the AGCM described in Numaguti et al. (1997), namely, CCSR/NIES/FRCGC AGCM5.7b. The horizontal resolution is set to T42, which corresponds to a grid size of approximately 2.8° longitude and latitude and the number of levels is set to 20 in the σ coordinate system where pressure at all heights is scaled with the surface pressure. USGS GTOPO30 is used to generate the surface elevation. The level 2 scheme of the turbulence closure model by Mellor and Yamada (1982) is used for sub-grid vertical fluxes of prognostic variables. A radiative transfer scheme (Nakajima et al., 2000) based on the two-stream discrete ordinate and k-distribution methods is employed. Other physical parameterisations include a prognostic Arakawa-Schubert cumulus scheme and a prognostic cloud water scheme for large-scale condensation (Le Treut and Li, 1991). Optical parameters for water cloud, ice cloud and five aerosol types - soil dust, black carbon, organic carbon, sulfate and sea salt - are included. Classification of aerosols is based on Spectral Radiation-Transport Model for Aerosol Species (SPRINTARS) (Takemura et al., 2000). Indirect effects of aerosols are considered for condensation in stratus clouds. Monthly aerosol mass and particle number concentration used in radiative processes are prescribed off-line by SPRINTARS.

The land-surface model used is Minimal Advanced Treatments of Surface Interaction and Runoff (MATSIRO) (Takata et al, 2003), whose horizontal resolution is the same as that of the atmospheric component. Here, water and heat exchange between the land surface and atmosphere is computed. Within the same model, runoff on the land is also calculated and passed over to a river routing model which transports the runoff water to the ocean model at river mouths. Prognostic variables include canopy water content, canopy temperature and soil moisture. Land-cover classification is derived from USGS GLCC (Global Land Cover Characterization). See Chan et al. (2011) for the present-day vegetation distribution.

The ocean component is basically version 3.4 of the CCSR Ocean Component Model (COCO) (Hasumi, 2000). The horizontal grid has 256x192 points so that each grid point is spaced equally at 1.40625° in the longitudinal direction. In the latitudinal direction, resolution is highest in the tropics (0.56°) and lowest at the polar regions (1.4°). The are 43 vertical levels, including 8 σ levels near the sea surface. Here, σ denotes a normalised geopotential height, with a value of 1 at the free surface and 0 at a fixed depth above which the σ coordinate system is applied. The Bering Strait throughflow is fully represented but the Hudson Bay and the Mediterranean Sea are treated as isolated lakes with heat and salinity exchanged with the open seas by a 2-way linear damping. A simple, vertical adjustment is applied, whereby unstable water columns are homogenised instantaneously. Vertical mixing of sea tracers and momentum use viscosity and diffusion coefficients calculated by Noh and Kim (1999). While there are no changes to the model as used in PlioMIP1, it should be noted that a larger Gent-McWilliams coefficient has been used in other more recent published work using MIROC4m (Obase and Abe-Ouchi, 2019, Sherriff-Tadano and Abe-Ouchi, 2020). This coefficient which refers to the horizontal diffusion of the isopycnal layer thickness is set to $300m^2$/s in the present study.

Sea ice concentration, thickness and horizontal velocities are calculated in the sea ice model. The equation of momentum includes an advection term, a Coriolis term, an acceleration term due to the slope of the sea surface, an internal stress term and an external forcing term derived from wind stress and ice-ocean drag. Upward longwave radiative flux is calculated according to the Stefan-Boltzmann law with an emissivity of 0.95. The albedo of bare ice surface is set to a constant value of 0.5, and that of snow-covered surface varies between 0.65 and 0.85, depending on the temperature. The air-sea/ice flux is calculated by taking into account downward shortwave and longwave radiative fluxes calculated in the atmospheric model.

## 3 Experimental design

The nomenclature used for the experiments follows that specified for PlioMIP2 (Haywood et al, 2016), that is, it takes the form $Ex^c$, where c is the concentration of $CO_2$ in ppm, and x can be any combination of the Pliocene boundary conditions. With present day boundary conditions only, x is null, otherwise it can be o (Pliocene orography, bathymetry, land-sea mask, lakes and soils combined) and/or i (Pliocene ice sheets).

Altogether, results from 8 experiments are included in this study. Firstly, there are the two core experiments, the Pre-Industrial ($E^{280}$) and a Pliocene time slice ($Eoi^{400}$), which all modelling groups participating in PlioMIP2 are expected to run. Secondly, with the Pre-Industrial set-up, atmospheric $CO_2$ levels are increased to 400ppm and double that of the Pre-Industrial ($E^{400}$ and $E^{560}$). Thirdly, at the Pliocene time slice, atmospheric $CO_2$ levels are changed to 280ppm, 350ppm and 450ppm ($Eoi^{280}$, $Eoi^{350}$ and $Eoi^{450}$). These two groups form part of the Tier 1 and 2 experiments. Fourthly, we include an experiment with all boundary conditions set to those of the mPWP in PlioMIP1 and name this Eplio1 as it is not formally part of PlioMIP2. In the core, Pre-Industrial experiment, atmospheric $CO_2$, $CH_4$ and $N_2O$ levels were initially set to those used for the MIROC4m experiments in PlioMIP1 (see Chan et al., 2011) for consistency, and these levels differed slightly to the specifications set in PlioMIP2. All the other experiments in this study subsequently followed the Pre-Industrial, including the double $CO_2$ experiment in which $CO_2$ is set to approximately 571ppm. In order that MIROC4m results may be compared systematically with data from other modelling groups in future studies, the 8 experiments are continued for a further 1,000 model years with the three greenhouse gas levels set to those specified in Haywood et al. (2016). Final $CO_2$ levels and boundary conditions for each experiment are listed in Table 1. Common to all experiments are the other greenhouse gases, astronomical parameters and the solar constant which are listed in Table 2.

## 3.1 Pre-Industrial ($E^{280}$) and increased $CO_2$ experiments ($E^{400}$ and $E^{560}$)

These 3 experiments with present day land topography were run previous to this study. The $CO_2$ level for the Pre-Industrial ($E^{280}$) is initially set to 285.431ppm, in accordance to previous MIROC4m experiments, and thus double the Pre-Industrial level is set to 570.862ppm. The model was integrated for 1220, 2000 and 2920 years for $E^{280}$, $E^{400}$ and $E^{560}$, respectively, using these old greenhouse gas levels. The time series of the AMOC and global temperatures for the last 1000

years are shown on the extreme left in Figure 2. For this study, these experiments are continued for another 1000 model years with the greenhouse gases changed to levels specified in PlioMIP2; the time series for these 1000 years are plotted on the right-hand side of Figure 2.

### 3.2 Core Pliocene (Eoi[400]) and related experiments with different $CO_2$ levels (Eoi[280], Eoi[350] and Eoi[450])

With greenhouse gas levels initially set to their previous values, the core experiment, Eoi[400], and the same experiment with Pre-Industrial $CO_2$ levels, Eoi[280], start from E[280] and the model is integrated for 3000 and 1500 years, respectively. At the end of 3000 years, Eoi[350] and Eoi[450] branch off Eoi[400]; the model for these two branches is integrated for 2000 years. Then as before, with greenhouse gas levels modified to PlioMIP2 values for these 4 experiments, the model is integrated for a further 1000 years.

The full, enhanced boundary conditions from Haywood et al. (2016) are employed, in particular, the Pliocene minus Modern topography anomaly, as shown in Figure 1, is applied to the existing MIROC4m land elevation. The largest reductions in surface elevation can be found in Greenland and parts of Antarctica, whereas the largest increases are located over North America and in the interior of Antarctica. Note that the Pliocene surface elevations used in PlioMIP1 and PlioMIP2 differ from one another (Supplementary Figure 1a). In addition to a lower surface elevation in the northern half of Greenland for PlioMIP2 due to a smaller ice sheet, higher surface elevation is found in southern Greenland, much of North America, except over the southern Rocky Mountains, northern Eurasia and the northern and central Andes Mountains. The land-sea mask is modified according to PlioMIP2 paleogeography. Modifications which did not exist in PlioMIP1 include the closure of the Bering Strait and the Canadian Arctic Archipelago Straits (CAAS), and the introduction of Pliocene lakes across Africa.

For consistency we apply the same vegetation distribution as that in the PlioMIP1 study by Chan et al. (2011), derived from Salzmann et al. (2008). The vegetation is extended in several regions where the land mask differs from that of PlioMIP1, for example, over an expanded Indonesia, over Beringia, resulting from a closed Bering Strait, and across parts of Greenland where the ice sheets are now specified to be smaller than that of PlioMIP1. In addition, several lakes have now been included on the African continent. The bathymetry is modified by the usual anomaly method, although a separate experiment was carried out without such modifications, resulting in no noticeable differences in the climate. Soil types are changed according to their texture.

### 3.3 Pliocene with PlioMIP1 boundary conditions (Eplio1)

For the lone experiment with PlioMIP1 boundary conditions and $CO_2$ level set to 405ppm, the model is initialised with Pre-Industrial conditions and integrated for 3000 years before the other greenhouse gas levels are changed to PlioMIP2 values for a further 1000 years. As in our original PlioMIP1 experiment (Chan et al., 2011), there are no modifications made to the bathymetry, soil types or lakes.

## 4 Results and discussion

Henceforth, we call all experiments with PlioMIP1 or PlioMIP2 boundary conditions (Eplio1 and Eoi$^{xxx}$) as simply "Pliocene experiments", irrespective of $CO_2$ level. Analyses are mostly based on the last 100 years of the model integration for each experiment and focus on the differences between the Pliocene experiments, especially the core, default Eoi$^{400}$ experiment, and the Pre-Industrial E$^{280}$ experiment. We also examine the Pliocene climate for a range of $CO_2$ levels and determine how the Pliocene experiments, in particular the core experiment, compares with proxy data for surface air and sea surface temperatures.

### 4.1 Surface air temperature

The changes in the annual mean surface air temperature (SAT) from Pre-Industrial values are shown in Figure 3. When only the $CO_2$ level is increased (Figure 3a), temperature increases are less extreme, compared to the other experiments. E$^{560}$-E$^{280}$ is not shown in this figure, but spatially, it resembles E$^{400}$-E$^{280}$. Temperature increases are slightly higher over land, with the smallest increases over central Africa and south-east Asia. Over the oceans, there are small regions, mainly over the Greenland Sea, where temperature changes are small, or even negative, and this is also noticeable to some extent in the other experiments. The largest temperature increases occur around the edge of Antarctica and across the Barents Sea. When only Pliocene vegetation, ice sheets, land configuration and elevation are introduced, and the $CO_2$ level is left unchanged, as in Eoi$^{280}$-E$^{280}$ (Figure 3c), the largest temperature increases can be found in the exact regions where ice sheets have been removed and land elevation is consequently lowered, that is, West Antarctica, the coastal regions of East Antarctica just to the south of Australia, and Greenland. Conversely, over the interior of the Antarctic continent, there are also locations where the temperature has decreased because the land elevation has increased while ice sheet is still present. Combining all Pliocene boundary conditions with increased $CO_2$, the spatial distribution resembles that of Eoi$^{280}$-E$^{280}$. The near-linearity of the combined effects of $CO_2$ and Pliocene land conditions is also reflected in the globally averaged temperature increase shown in Table 3. This increase, the global SAT anomaly, for E$^{400}$ and Eoi$^{280}$ is 2.0°C and 1.1°C, respectively, and that for Eoi$^{400}$ is 3.1°C, which corresponds to a near-zero residual from non-linear effects, similar to the findings of Kamae et al. (2016) who further decomposed the Pliocene land conditions into ice sheets and all other effects. Others find that the combined effects are less than the sum of the individual effects (Hunter et al., 2019; Chandan and Peltier, 2018).

The overall temperature increase in Eoi$^{400}$ is not as large as that in the PlioMIP1 experiment (Table 3), as in the case with the models HadCM3 (Bragg et al., 2012; Hunter et al., 2019), NorESM-L (Li et al., 2020) and COSMOS (Samakinwa et al., 2020), with the first model and MIROC4m both giving a PlioMIP2-PlioMIP1 global SAT difference of 0.4°C. Other models, as noted by Li et al (2020), show the opposite result, ie. larger warmer in PlioMIP2. In our results, overall, the PlioMIP1 temperatures lie between those of PlioMIP2 with $CO_2$ set to 400 and 450ppm. PlioMIP2 temperature increases in most regions of the northern high latitudes and the tropics are smaller and this may be at least partly a result of the increased elevation across most of the northern hemisphere, especially over North America, as the difference between Eoi$^{400}$ and Eplio1

SAT follows closely that of the elevation difference (Supplementary Figures 1a and 1b). Conversely, SAT in Eoi$^{400}$ is much higher than that in Eplio1 across West Antarctica, northern Greenland, the southern Rocky Mountains and the Near East, the last three of which also exhibit the same SAT differences in NorESM-L (Figure 9a of Li et al., 2020).

The globally averaged SAT from PlioMIP has been used to estimate the Earth system sensitivity (ESS), which, unlike the climate sensitivity (CS), takes into account feedbacks operating over longer timescales (Lunt et al., 2009). From E$^{280}$ and E$^{560}$, CS is estimated to be 3.9, and using Eplio1 and Eoi$^{400}$, ESS is estimated to be 6.6 and 6.0, respectively, and the ESS/CS ratio 1.7 and 1.5, respectively. All these values compare quite well with the PlioMIP2 multi-model mean values (Haywood et al., 2020).

Figure 4 shows the zonal mean surface air temperature increase from E$^{280}$. When only $CO_2$ is changed, a small but gradual and near-linear increase in temperature anomaly starting from the southern mid-latitudes to the northern mid-latitudes is evident (see the red and purple lines between 45°S and 45°N). The larger anomaly in the northern half is simply a consequence of the larger land area there. Increase in $CO_2$ is also accompanied by polar amplification of the warming. For all other experiments, the increase in temperature anomaly from the tropics to the northern mid-latitudes is more pronounced. In the northern polar region, peak warming is seen at around 75°N for all experiments. However, in the southern polar region, peak warming shifts from 65°S to 75°S with the inclusion of Pliocene land elevation and reduced ice sheets. For the Pliocene experiments, Eoi$^{350}$, Eoi$^{400}$ and Eoi$^{450}$, the increase in zonal temperature anomaly with an increase of 50ppm in $CO_2$ is limited to less than 1°C at low and mid-latitudes, and at most 1.5°C at the polar regions. This increase is relatively small in comparison to results from models like IPSL-CM5A2 which shows a fairly small uniform change from Eoi350 to Eoi400, except at the northern high latitudes where there is a sharp change of up to 2.5°C (Figure 11c of Tan et al., 2020).

The seasonal SAT anomalies for Eoi$^{400}$ are shown in Figure 5. There is little seasonal change over north Africa and much of the oceans. Throughout the year, temperature increases over Greenland remain large. However, there are distinct seasonal changes at high latitudes elsewhere, for example, the small temperature increase in the Arctic region during the summer, followed immediately by the extremely large temperature increase during the autumn. In the summer, there is very little sea ice in the Arctic in Eoi$^{400}$ and so the ocean warms up more from incoming insolation until the SST reaches a maximum. As the summer ends, heat from the ocean is released back into the atmosphere. Since there is very little sea ice, more heat can be released, explaining the higher SAT in the Arctic during September to November. This was also seen in the study by Zheng et al (2019). The Hudson Bay shows a large temperature reduction during winter because it has been replaced by land which cannot stabilise the surface air temperature as much as water can (Hunter et al., 2019). Conversely, by the same reasoning, a large temperature increase is seen over the Hudson Bay during the summer. Another region where the temperature reduces is the zonal strip in Africa at latitude 15°N during the summer. This is similar to what was seen in the early work of Chandler et al. (1996) who attributed this feature to a weakening of the Hadley circulation. Both the summer cooling over this part of Africa and the Indian subcontinent are also clearly seen in Hunter et al. (2019).

## 4.2 Sea surface temperature

Annual mean sea surface temperature (SST) anomalies, shown in Figure 6, are similar to SAT anomalies. With only an increase in $CO_2$ (Figure 6a), there is a uniform increase in SST, except in the eastern Pacific and Indian Ocean sectors of the Southern Ocean, in the north-west Pacific Ocean and in the Greenland Sea. The inclusion of all other Pliocene boundary conditions leads to more extreme SST increases over the northern mid to high latitudes, as well as the western half of the Indian Ocean. The Greenland Sea is the only region where SST decreases whether by increasing $CO_2$ to 400ppm (Figure 6a) or by introducing Pliocene boundary conditions (Figure 6c) and this cooling persists in all the Pliocene experiments. The global SST for Eoi$^{400}$ is 19.0°C (Table 3), slightly lower than that for the PlioMIP1 experiment (19.2°C).

It is later shown in Supplementary Figure 1(c) and Figure 15(b) that SST is much lower in Eoi$^{400}$ than in Eplio1 in the Barents Sea, along the north-west Pacific coastal regions, but much higher in the Labrador Sea, the eastern North Atlantic Ocean and parts of the Southern Ocean, which follows the SAT in the coastal regions of Antarctica. The difference in the SST of the Barents Sea in the two experiments could be explained by the higher land elevation over northern Europe in PlioMIP2 which would lead to SAT lower than those in Eplio1 in the same way higher land elevation over North America leads to SAT there being lower in Eoi$^{400}$ than in Eplio1. The slightly warmer SST in the Labrador Sea results from the closure of the CAAS in the PlioMIP2 paleogeography since cooler waters flow southward from the Arctic Ocean to the Labrador Sea via the open CAAS in Eplio1. Not only does MIROC4m and NorESM-L show a similar warming in PlioMIP2 SST in the Labrador Sea (Figure 9b of Li et al., 2020), but both models also show slightly higher SST in PlioMIP2 than in PlioMIP1 in coastal areas off Mexico, Chile, northwest Africa and southwest Africa, suggesting subtle differences in coastal upwelling.

## 4.3 Precipitation

For the Pliocene experiments, the fractional increase in the annual mean precipitation (Figure 7) is greatest over most of Antarctica, northern Africa, the Indian Ocean, central Asia and at the northern high latitudes, in particular, across Greenland and northern Canada. Consequently, the sea surface salinity (not shown) in the Indian Ocean reduces by a large amount. Conversely, the precipitation decreases most noticeably over the oceans elsewhere, in the tropics and subtropics, in addition to the Greenland Sea where SAT decreases. Over land, the precipitation decreases across south-east North America and southern Africa. As the $CO_2$ level is increased, these changes are accentuated and the globally averaged precipitation increases (column 8, Table 3). The opposing changes across northern Africa and the northern South Atlantic Ocean is indicative of a northward shift of the Intertropical Convergence Zone (ITCZ). It can be seen from a comparison of E$^{400}$-E$^{280}$ (Figure 7a) and Eoi$^{280}$-E$^{280}$ (Figure 7c) that most of these changes in precipitation occur as a result of the Pliocene boundary conditions, rather than the change in $CO_2$. The increase in $CO_2$ is responsible for a moderate increase in precipitation over the whole of Antarctica and for a small decrease over northern Africa, opposite to the effect of the Pliocene boundary conditions. As with SAT, the globally averaged precipitation in Eoi$^{400}$ is larger than that in E$^{280}$, with $CO_2$ contributing about two-thirds of the total increase.

The spatial changes in the PlioMIP1 experiment are similar to those in Eoi[400], with the global changes in precipitation also being similar (Table 3). However, there are small differences in the amount of precipitation change (Supplementary Figure 1d). In many areas, precipitation increases with decreasing surface elevation and vice versa, for example, over Greenland and Antarctica. Eoi[400] precipitation is lower in the Indian Ocean, west of Australia, while it is higher in northern Africa and the tropical Atlantic. This may be related to differences in the ITCZ.

## 4.4 Sea ice and ocean mixed layer depth

The total sea ice area in the polar regions is depicted in Figure 8. In the Arctic, during March when the sea ice area is at its greatest, there is a gradual decrease in area as $CO_2$ is increased, whether Pre-Industrial or Pliocene boundary conditions are used. The sea ice area in Eplio1 is actually larger than any of the other Pliocene experiments because the land area around the Arctic is smaller in Eplio1, as can be seen later in Figure 9 – the Bering Strait is open, the Hudson Bay is still set as open water and the Labrador Sea is larger and connected to the Arctic Ocean. During September, when the sea ice area is at its smallest, a similar trend is observed. The Arctic Ocean is ice-free in Eplio1, Eoi[400] and Eoi[450]. In the Antarctic, during September, the decrease in sea ice area as $CO_2$ increases is much more drastic. The Antarctic sea ice area when the $CO_2$ level is doubled is also much lower than that of any of the Pliocene experiments. Unlike the Arctic, the sea ice area in Eplio1 and Eoi[400] are similar as the Antarctic coastlines in the two cases do not differ by much. The same behaviour is observed in March, although unlike in the Arctic, the Antarctic is never ice-free. However, in both polar regions, the boundary of sea ice extent in Eoi[400] and Eplio1 are similar (Figures 9 and 10).

The sea ice extent in the Arctic and the mixed layer depth (MLD) in the surrounding regions during March are shown in Figure 9. The definition of the MLD follows that of Oka et al. (2006), i.e. the depth at which $\sigma_\theta$, the potential density anomaly, differs from the surface value by 0.1. The sea ice extent here is largely unaffected by the Pliocene boundary conditions alone and only recedes near the north-west Pacific coastline and in the Barents Sea as $CO_2$ is increased. In E[280], the MLD is large everywhere south of the sea ice extent in the North Atlantic, including the Labrador Sea and the Norwegian Sea, and to a lesser extent west of the British Isles. The Labrador Sea responds with a larger MLD, spread over a greater area directly south of Greenland to Pliocene boundary conditions and increasing $CO_2$. when applied separately. However, when both Pliocene boundary conditions and changes in $CO_2$ are applied, as in Figures 9(e-h), it can be seen that $CO_2$ has an opposing effect on the MLD, suggesting that deepwater formation in the Labrador Sea is greater in Eoi[280] than in Eoi[450]. In the Eplio1 experiment, lower SST in the Labrador Sea, as discussed previously, also contributes to greater deepwater formation so that the MLD is more comparable to that of Eoi[280]. In the Norwegian Sea, the areal extent of deep MLD is similar in all experiments, but only E[280] shows the maximum MLD of about 3000m, located at the northernmost ice-free region, at latitude 80°N. In all experiments with higher $CO_2$, the sea ice in the northeast of the Norwegian Sea retreats and exposes the Barents Sea, but the extent of the deepwater formation region in the Norwegian Sea remains unchanged. In the region to the west of the British Isles, regardless of $CO_2$ level, no change is seen when PlioMIP2 boundary conditions are applied. On the hand, there is little deepwater formation there in Eplio1 or when $CO_2$ level is 400ppm or higher with present day boundary conditions.

The sea ice extent and MLD in the Antarctic during September are shown in Figure 10. With present day boundary conditions ($E^{280}$, $E^{400}$ and $E^{560}$), the MLD is extremely large in the Atlantic sector of the Southern Ocean, even below parts of the sea ice in $E^{280}$, suggesting the formation of dense water due to brine rejection. In the Pliocene experiments, this large MLD is absent, while in the other region where the MLD is large with present day boundary conditions, i.e. the eastern South Pacific, it is reduced.

## 4.5 Atlantic meridional overturning circulation

The Atlantic meridional overturning circulation (AMOC) is shown in Figure 11 and the AMOC index, defined as the maximum streamfunction value, in the last column of Table 3. Increasing the $CO_2$ level alone has a tendency to weaken the AMOC. As the overturning cell becomes shallower, the underlying Antarctic Bottom Water extends further northward. In addition, the anti-clockwise overturning cell north of 65°N strengthens, contributing to increased convection and deepwater formation in the Labrador Sea, as indicated by the MLD in Figure 9. A comparison of $Eoi^{280}$ with $E^{280}$ shows that similar changes occur when only the Pliocene boundary conditions are applied, except that the AMOC index increases by nearly 0.7Sv, despite shoaling of the AMOC. Thus, increasing $CO_2$ and applying Pliocene boundary conditions have an opposite effect on the AMOC index, but up to 450ppm $CO_2$, the indices for all the Pliocene experiments are still greater than that for $E^{280}$. Note that the degree of weakening in the AMOC as $CO_2$ increases seems to be highly dependent on the boundary conditions. With present day boundary conditions, from $E^{280}$ to $E^{400}$, the AMOC index decreases by 0.8Sv, whereas in the corresponding Pliocene experiments, $Eoi^{280}$ and $Eoi^{400}$, there is only a decrease of approximately 0.2Sv (Table 3).

A comparison of $Eoi^{400}$ and Eplio1 shows that while the AMOC cell extends to similar depths and the circulation in the other two cells change little, the AMOC index with PlioMIP2 boundary conditions is larger than that with PlioMIP1, 20.0Sv versus 17.8Sv. While we have not performed specific sensitivity experiments to see what difference between the PlioMIP1 and PlioMIP2 boundary conditions is exactly responsible for this difference in the AMOC index, we did perform some (not shown) with a Pre-Industrial background climate and looked at the effects of closing the Bering Strait and the CAAS closed. We find that, as in Pliocene studies by Otto-Bliesner et al. (2017), closure of the Bering Strait, irrespective of the state of the CAAS, leads to a stronger AMOC. Closing the Bering Strait inhibits the transport of freshwater from the North Pacific Ocean via the Arctic Ocean and increases the AMOC index by 1-2Sv in the MIROC4m experiments, and so it likely explains the difference between $Eoi^{400}$ and Eplio1. This increase is sufficient to give a stronger Pliocene AMOC when compared to the Pre-Industrial, a result which is opposite to when using PlioMIP1 boundary conditions and which is consistent with recent PlioMIP2 simulations with other models (e.g. Chandan and Peltier, 2017; Tan et al., 2019; Hunter et al., 2019; Li et al., 2020; Feng et al., 2020).

## 4.6 Meridional heat transport

Figure 12(a) shows the meridional ocean heat transport in the Atlantic Ocean for $E^{280}$ and the Pliocene experiments. The transport peaks at around 0.7PW, near latitude 15°N. Figure 12(b) shows the difference between the same heat transport

in the Pliocene experiments and $E^{280}$. In all the Pliocene experiments, there is a reduction in the heat transport at all latitudes, with the greatest reduction at the equator for experiments using PlioMIP2 boundary conditions. For these four PlioMIP2 Pliocene experiments, there does not appear to be any general trend as the $CO_2$ level is increased. For example, $Eoi^{450}$ has a significantly lower heat transport compared to the others south of the equator, whereas at the northern low to mid-latitudes, $Eoi^{450}$ values are rather close to those of $Eoi^{400}$ and it is the $Eoi^{280}$ values that stray from the rest. It can also be seen that there is a marked difference between the heat transport in Eplio1 and those using PlioMIP2 boundary conditions. At all latitudes south of 30°N, heat transport in Eplio1 is about 0.1PW lower than that of $E^{280}$. Irrespective of $CO_2$ level, this larger difference can mostly be attributed to the differences between the PlioMIP1 and PlioMIP2 boundary conditions, as can the difference between the AMOC index of Eplio1 and those of the PlioMIP2 Pliocene experiments. However, as noted in the latter experiments, the northward heat transport decreases even though the AMOC index increases, albeit to a relatively small degree. While these two properties are commonly thought of as being positively correlated to each other, Sévellec and Fedorov (2016) show that this is not always the case. Using NorESM-L, $Eoi^{400}$ exhibits the same features, ie. reduced meridional heat transport in the Atlantic Ocean but a stronger AMOC (Li et al., 2020), whereas its sister model, NorESM1-F, as well as many other models, show both increased heat transport and a stronger AMOC (Kamae et al., 2016; Chandan and Peltier, 2017; Tan et al., 2020).

Figure 13 shows the total meridional heat transport in a similar way, but for the whole climate system, combining atmosphere and ocean. The absolute values, in the Figure 13(a), confirm that, in general, heat is transported northwards in the northern hemisphere and southwards in the southern hemisphere. In some studies, the total heat transport in both hemispheres in $Eoi^{400}$ either decreases (Chandan and Peltier, 2017) or increases (Feng et al., 2020). In contrast, for all Pliocene experiments in this study, northward heat transport is reduced in the northern hemisphere, while southward heat transport is increased in the southern hemisphere (Figure 13b). This appears to be more akin to the results from IPSL-CM5A, as seen in Figure 6(a) of Tan et al. (2020), although the anomalies are not shown in their study explicitly. How these individual experiments differ from one another depends on the hemisphere. Firstly, with PlioMIP2 boundary conditions, as the $CO_2$ level is increased, there is a clear, monotonous trend with the magnitude of the northward heat transport anomaly reducing in the northern hemisphere. The opposite trend is seen in the southern hemisphere as southward heat transport increases with $CO_2$. $Eoi^{400}$ represents a half-way mark whereby the bold green line in Figure 13(b) is roughly symmetric across the equator. Secondly, Eplio1 aligns very closely with $Eoi^{400}$ in the southern hemisphere, whereas in the northern hemisphere, Eplio1 sets itself apart from the other Pliocene experiments and its heat transport reduces much less. Finally, we note that the largest anomalies occur at low latitudes, a characteristic not evident in the studies mentioned above which tend to show greatest anomalies at the mid-latitudes where the absolute values are at their peak.

### 4.7 Comparison with surface air temperature proxy data

In Figure 14, the annual mean SAT from the Pliocene experiments and $E^{280}$ are compared with SAT estimates from vegetation reconstructions at marine sites near land, as compiled by Salzmann et al. (2013). The red symbols refer to the

PlioMIP2 Pliocene experiments, with the lower and higher horizontal red lines referring to 350ppm and 450ppm $CO_2$, respectively. For most locations, SAT from these experiments show the correct tendency and agree better with the proxy data than $E^{280}$. Locations where the Pliocene values do not match proxy data as well as $E^{280}$ values include Rio Maior, Yorktown and Pinecrest, where the Pliocene values are too high and proxy data values are very close to $E^{280}$ values. However, the biggest

discrepancies are seen in the Charan Basin and Lake Baikal, but even here, the Pliocene values match proxy data better than $E^{280}$ values. Moreover, the Pliocene model values at these two locations are very close to the CCSM4 model results in Table 4 of Chandan and Peltier (2017). There are many locations where the Pliocene SAT, in particular that of $Eoi^{400}$, fall within the proxy data uncertainty range, which, in turn, is significantly higher than the $E^{280}$ values, eg Andalucia G1, Habibas and Nador. Eplio1 values, for the most part, are similar to $Eoi^{400}$ or $Eoi^{450}$ values. Thus, in general, the Pliocene experiments show good

agreement with the proxy data, but there does not appear to be any particular value of $CO_2$ which gives the best fit.

A similar comparison between model SAT and estimates from terrestrial vegetation data in the same study is shown in Figure 15. The sites in the figure are listed according to latitude, following Table S3a of Salzmann et al. (2013), with those on the left being located at northern high latitudes. At the most northerly sites (1-10), most of which are located in Alaska and Siberia, model SAT increases by at least 5°C, consistent with the warming suggested by the vegetation data, but the degree

of warming is insufficient. As noted before, Eplio1 SAT is higher than SAT from the other Pliocene experiments at northern high latitudes. At sites 11-20, located at latitudes 56°N-47°N, the degree of warming is, in most cases, sufficiently high such that Pliocene SAT at higher levels of $CO_2$ fall within the range of SAT derived from vegetation data. At the remaining sites, further south, Pliocene warming is also seen, although there are some sites (e.g. 26-28) where Pre-Industrial SAT is already equal to or higher than the proxy value.

**4.8 Comparison with sea surface temperature proxy data**

PRISM SST proxy data from a variety of marine sources have formed an integral part of PlioMIP, as boundary conditions for AGCM experiments in the first phase and, more importantly, as part of data-model comparison. Figure 16(a) compares the annual mean $Eoi^{450}$-$E^{280}$ SST anomalies with the corresponding anomalies from PRISM3 sites where the colour refers to the difference between these two anomalies, with yellow, orange and red meaning model SST anomalies are greater.

The symbols show whether the proxy anomalies suggest a warmer Pliocene (triangles) or a cooler Pliocene (circles). Only a few PRISM3 sites show a cooler Pliocene, at low latitudes, and $Eoi^{400}$ does not replicate this cooling, especially in the Indian Ocean. In general, there is good agreement in the Southern Ocean where proxy data suggest a warmer Pliocene. In the northern hemisphere, where PRISM3 sites also suggest a warmer Pliocene, $Eoi^{400}$ overestimates the SST increase by up to 3°C in parts of the southern North Atlantic. As with model results from PlioMIP1, the large degree of warming in the northern North

Atlantic and Greenland Sea is not replicated (blue triangles). A comparison of $Eoi^{400}$ and Eplio1 SST anomalies is shown by the colours in Figure 16(b). SST anomalies in $Eoi^{400}$ are larger mainly in the Indian and Pacific sectors of the Southern Ocean, the eastern Atlantic Ocean and the Labrador Sea. While the SST anomalies in much of the northern North Atlantic and the

Barents Sea are lower in Eoi400 compared to Eplio1, the Eoi$^{400}$ values at the PRISM3 sites are actually higher, and thus agree better with the proxy data, as shown by the yellow triangles.

We also include a similar comparison between model results and the newer PRISM4 proxy SST data sets (Foley and Dowsett, 2019) in Figure 17. The degree of warming in Eoi$^{400}$ is much less than that suggested by PRISM4 proxy data in the Atlantic sites, in particular, at northern high latitudes again, and also near southern Africa. Slightly higher warming in Eoi$^{400}$ is generally seen in the other sites, mostly located at low latitudes, but especially in the Caribbean Sea, which is not so evident in the PRISM3 comparison (Figure 16a). The PRISM4 data used here refer to the broader 30ka interval, but alternative data

for a 10ka interval give the same conclusions. Qualitatively speaking, at least, these results are similar to those obtained from the multi-model mean in Figure 8(c) of Haywood et al (2020).

        For a more global sense of how the various Pliocene experiments compare with the proxy data, we refer back to the global PRISM3 SST field which was used as boundary conditions for AGCM experiments in PlioMIP1 and list the spatially and annually averaged model-data difference in Table 4. An important caveat here is that this global SST field is reconstructed

from data at a finite number of sites for February and August using interpolation and extrapolation, and that there are regions where data are sparse (Dowsett et al., 1999). Comparing Eplio1 and Eoi$^{400}$, we see that the former matches the proxy data better at northern high latitudes, and even better at southern high latitudes (a difference of only 0.04°C). On the other hand, the latter shows a smaller discrepancy at the tropics and low latitudes (30°S-30°N) where the larger surface area means that, globally speaking, Eoi$^{400}$ gives a better fit (a difference of 0.76°C). Next, a comparison of Eoi$^{350}$, Eoi$^{400}$ and Eoi$^{450}$ shows that,

while there is a trend in the model-data difference as $CO_2$ is reduced, there is no particular level at which the difference is small at the three latitudinal ranges in Table 4. Since the global difference is determined more by the low latitudes, Eoi$^{350}$ gives the best global fit (a difference of 0.14°C). It is worth noting that not only does Eoi$^{280}$ give the best fit at low latitudes, but the discrepancy is smaller than that for Eoi$^{350}$ at northern high latitudes, despite the increasing trend in discrepancy from Eoi$^{450}$ to Eoi$^{350}$.

For reference, the discrepancies between the model and proxy SST anomalies at the PRISM3 sites for different $CO_2$ levels are shown in Supplementary Figure 2. Similarly, comparisons using PRISM4 data for different $CO_2$ levels are shown in Supplementary Figure 3. There is no spatial reconstruction to accompany the data from PRISM4 sites, and so comparisons are based solely on these sites. Lower values of $CO_2$ tend to give better agreement at low latitudes, except to the west of western Africa where Eoi$^{450}$ shows good agreement. However, even in Eoi$^{450}$, Pliocene warming in the Greenland and

Norwegian Seas, the northern North Atlantic Ocean and the coastal region near southern Africa is much lower than that suggested by PRISM4 data (blue triangles).

        More recent studies by Tierney et al (2019) have highlighted uncertainties in tropical SST estimates and showed good agreement between their reduced space reconstruction and a Pliocene simulation run with CESM1. We carry out SST comparison (Figure 18) between our Pliocene experimental results and their reconstruction, limited to the Pacific Ocean, using

alkenone proxy data and a probabilistic approach. In the Tropical Pacific, this proxy reconstruction exhibits higher temperature anomalies than those from PRISM3, leading to optimal agreement with Eoi400 (Figure 18c), which was warmer than PRISM3

data by 1-3°C in that same region (not shown).  Although the proxy reconstruction shows enhanced warming in the eastern equatorial Pacific and in the northern subtropical Pacific, this warming is underestimated by 1-3°C, compared to PRISM3. This is most evident in the northwest Pacific where the reconstruction has more limited data, and warming is weaker than that in all our Pliocene experiments.

## 5 Summary and conclusions

In the present study, we have shown some basic results from the core PlioMIP2 experiment using the MIROC4m AOGCM, compared them to results from both the PlioMIP1 experiment and some other Tier 1 and Tier 2 PlioMIP2 sensitivity experiments.  Additionally, we have evaluated the consistency between these experimental results and some temperature proxy data from marine and terrestrial sources.

For the core experiment, PlioMIP2 boundary conditions produce a global temperature increase smaller than that with PlioMIP1 and it can be assumed that the $CO_2$ level has little effect as it differs only slightly in the two phases of PlioMIP.  The difference in the results from these two experiments is not uniform as greater warming is seen in PlioMIP2 in parts of the northern high latitudes and of the Southern Ocean.  Moreover, PlioMIP2 SSTs actually reconcile better with proxy-derived values at PRISM3 sites in the northern North Atlantic and Greenland Sea, albeit to a very small degree, although the large discord in the northern North Atlantic Ocean SSTs still remains.  For SAT, both PlioMIP1 and PlioMIP2 values show fairly good agreement with proxy data from paleobotanical sites, although comparisons at the northern high latitude sites highlight the weaker polar amplification in model results.  Northern polar amplification is slightly less in PlioMIP2, but nonetheless, zonal SAT increases are more than double that of the low latitudes.  Our sensitivity experiments have only distinguished between the two forcings from $CO_2$ and Pliocene boundary conditions and we have not considered the effects from the ice sheets, orography and vegetation separately.  $CO_2$ accounts for two-thirds of the total surface air temperature and precipitation increase.  Unlike PlioMIP1, the AMOC in PlioMIP2 is stronger compared to the Pre-Industrial for MIROC4m, which is in line with other model results published so far in PlioMIP2.  The strengthening of the AMOC from PlioMIP1 to PlioMIP2 is tied to the closure of the Bering Strait.

We have also looked at the mid-Pliocene climate for a range of $CO_2$ values.  From these $CO_2$ sensitivity experiments, we find that, not only does the AMOC strength decrease with increasing $CO_2$, but that this dependency on $CO_2$ is weaker when Pliocene boundary conditions are applied.  While other expected trends are seen, such as the increase in global temperature and precipitation with $CO_2$, of much importance is the comparison with proxy-derived data. Mismatches between Eoi[400] and proxy-derived SSTs at low and high latitudes are of the opposite sign, and data in both regions cannot be simultaneously reconciled simply by changing the $CO_2$ value.  From a global perspective, a value below 400ppm leads to a better overall fit with PRISM3 data. However, the warming at many of the newer PRISM4 proxy data sites is too high to be reconciled with model data, even at higher $CO_2$ levels.  In the tropical Pacific, more recent reconstructions suggest that Eoi[400] does not overestimate the SST, as implied by PRISM3 data, thus reducing the global discrepancy.  These results underscore firstly, the

difficulties in assessing model results with proxy evidence because of different regional discrepancies and uncertainties in proxy estimates, and secondly, a continuing requirement for additional proxy estimates.

Our experiments have not included dynamical vegetation but previous research employing the same model coupled to a dynamic global vegetation model have shown that such vegetation feedback can amplify both warming in the mid-Holocene (O'ishi and Abe-Ouchi, 2011) and cooling in the cold climate of the LGM (O'ishi and Abe-Ouchi, 2013), especially at northern high latitudes. Subsequent Pliocene-related MIROC4m climate simulations will include a configuration with dynamical vegetation. Other recent studies have incorporated the effects of orbital forcing in simulations of the mPWP, with modelled climate, ice sheets and vegetation exhibiting strong regional variations associated with orbital parameters, whether as time-dependent forcing in transient simulations (Willeit et al., 2013) or fixed to minimum or maximum forcings (Dolan et al., 2011; Feng et al., 2017). While using present-day orbital parameters for the KM5c interglacial peak appears valid, at least with fixed vegetation (Hunter et al., 2019), an investigation of the mPWP as a whole necessitates more realistic orbital parameters, even when restricted to interglacial peaks (Prescott et al., 2018). This should be borne in mind when considering paleoclimate modelling experiments such as those for the mPWP.

**Data availability**

Data files containing the PlioMIP2 boundary conditions are directly accessible from the USGS PlioMIP2 website, https://geology.er.usgs.gov/egpsc/prism/7_pliomip2.html. Data for most experiments in this study are available on the PlioMIP2 data repository at the University of Leeds, sftp://see-gw-01.leeds.ac.uk. Request for access should be directed to Alan M. Haywood. For all other data, readers are asked to contact the lead author, Wing-Le Chan (wlchan@aori.u-tokyo.ac.jp).

**Author contributions**

Both authors contributed to the writing of the paper and discussions. Wing-Le Chan set up and carried out the experiments, wrote the first draft of the manuscript and prepared all the figures.

**Competing interests**

The authors declare that they have no conflict of interest.

**Acknowledgements**

The authors acknowledge funding from JSPS KAKENHI grant 17H06104 and MEXT KAKENHI grant 17H06323, and
JAMSTEC for use of the Earth Simulator supercomputer. This manuscript benefitted greatly from the comments of two
anonymous reviewers. The authors would like to thank them and the editor, Aisling M. Dolan.

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

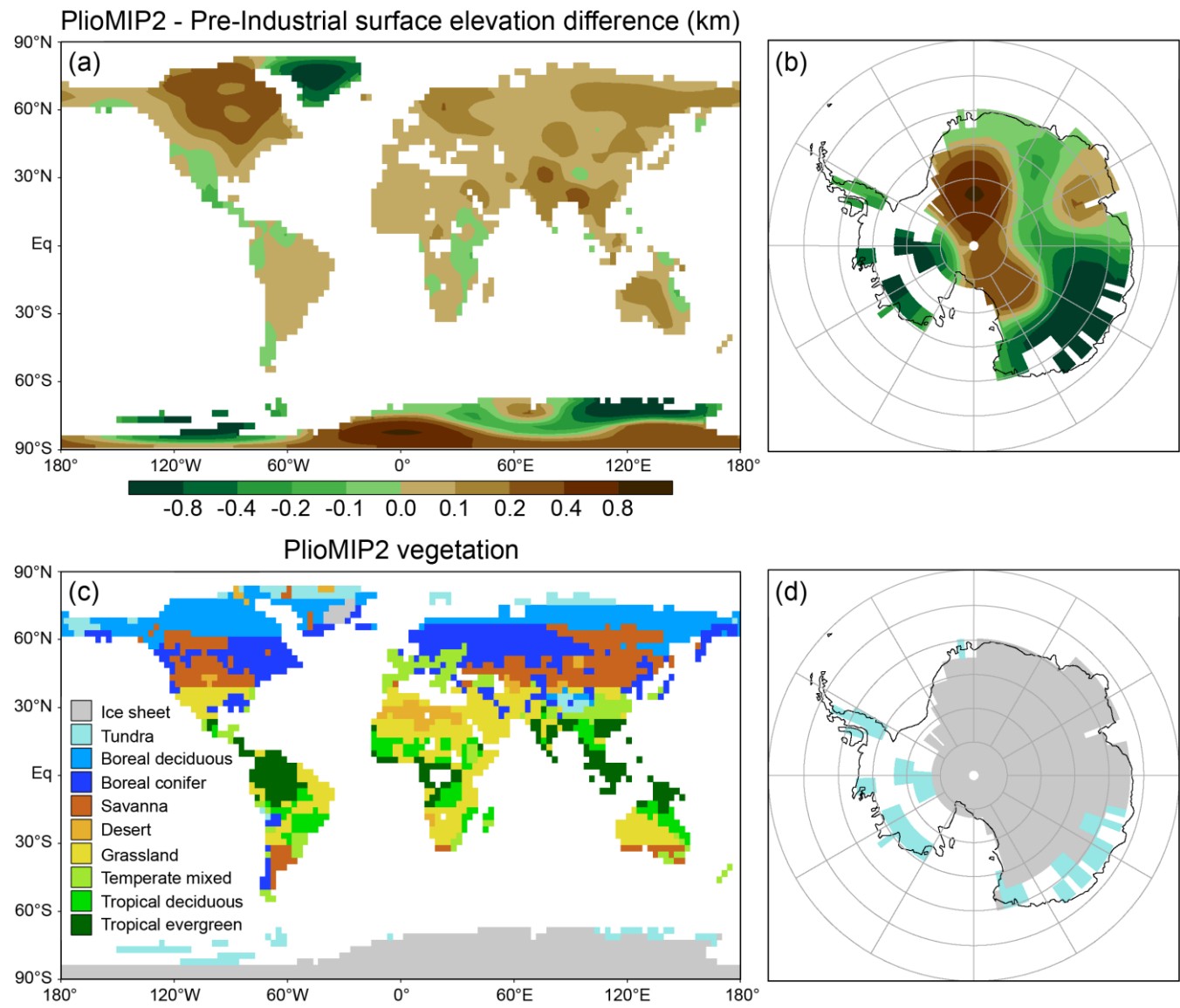

**Figure 1: (a) The difference between land elevation in the Pliocene and present day experiments, and (b) the vegetation specified for Pliocene experiments on the MIROC4m grid.**

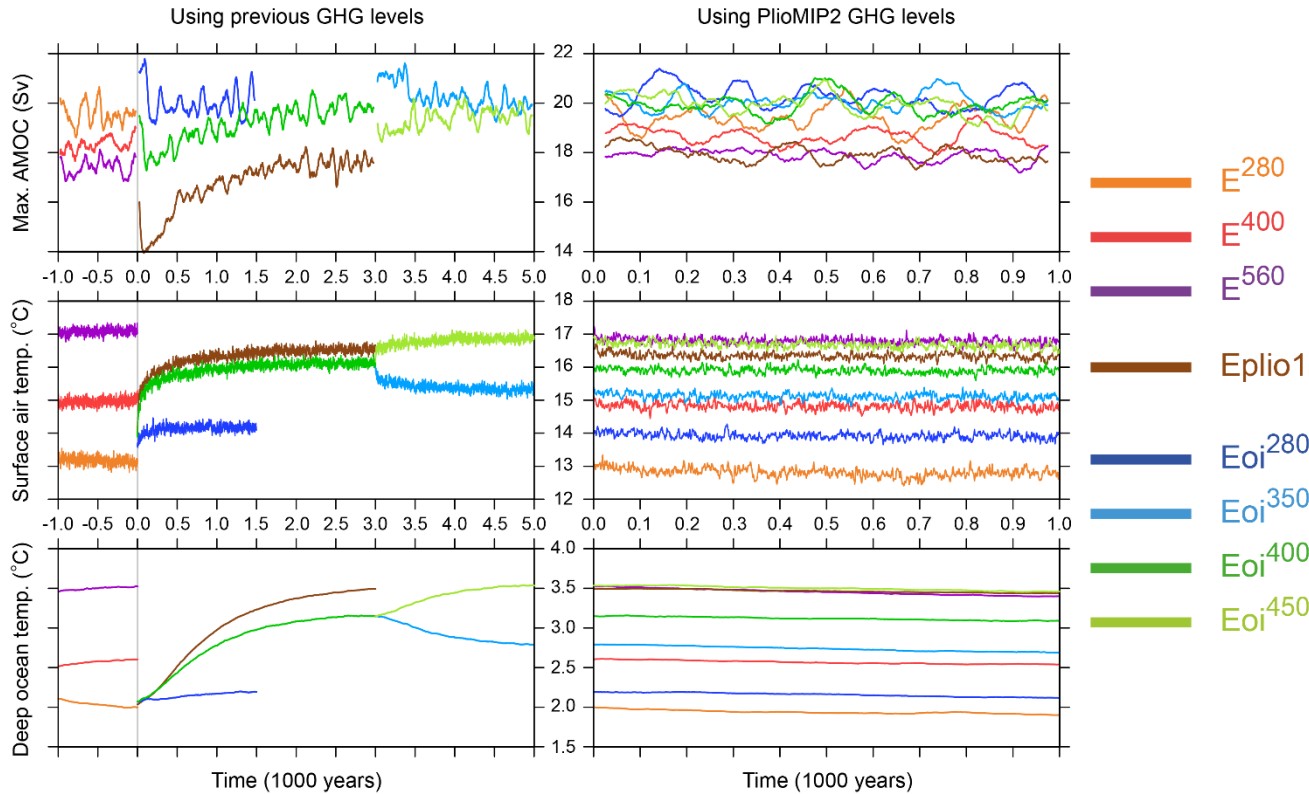

**Figure 2: Time series of the AMOC index (top), globally averaged surface air temperature (middle) and global ocean temperature below depths of 1900m (bottom). A 51-year moving average has been applied to the AMOC index time series. The subfigures on the left depict the initial stages of the experiments during which greenhouse gas levels are set to previous, PlioMIP1 values, while the subfigures on the right depict the final 1000 years of the integration during which greenhouse gas levels are consistent with those specified in PlioMIP2.**


# Annual mean surface air temperature anomaly (°C)

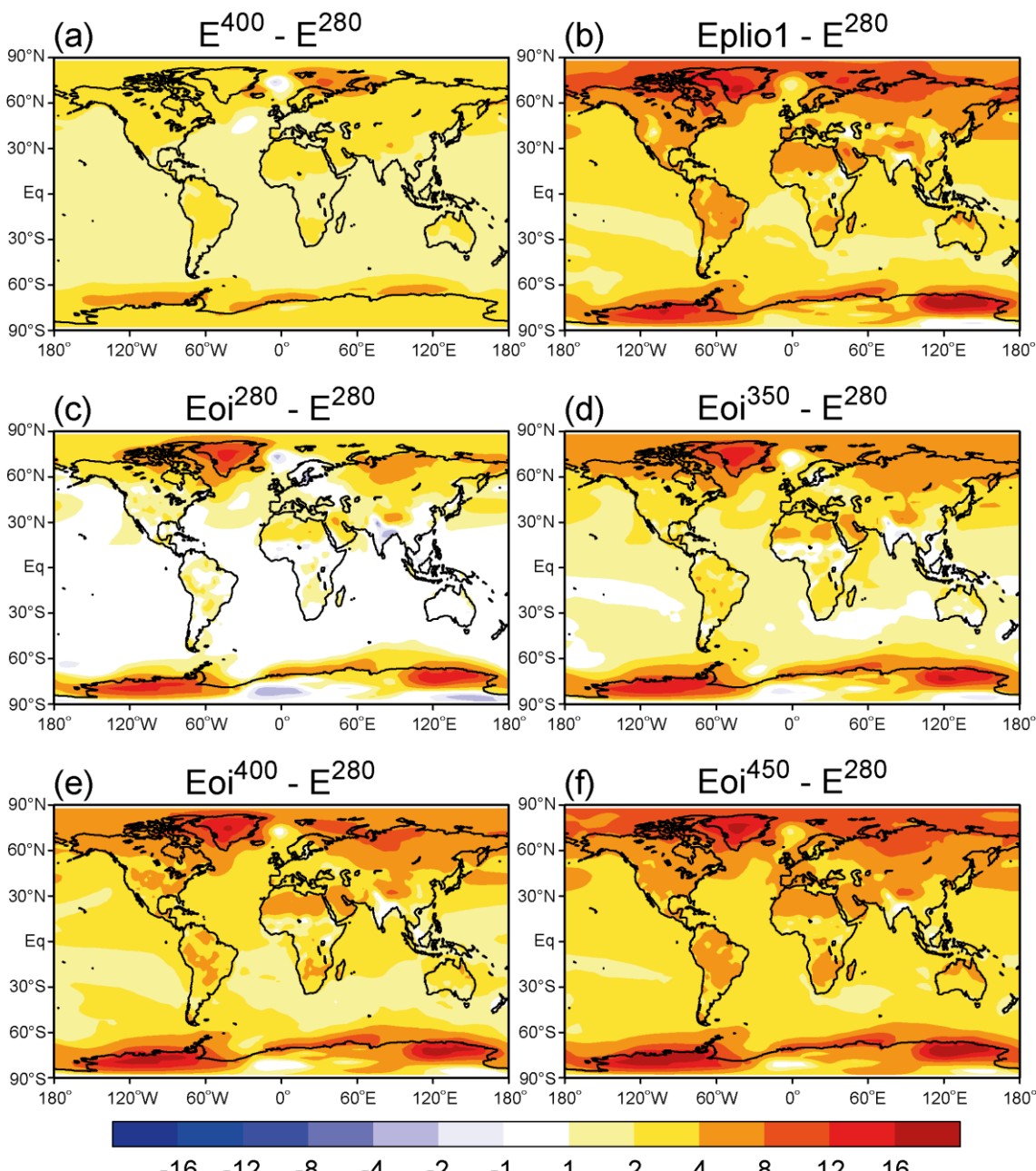

Figure 3: Annual mean surface air temperature anomaly between six experiments and $E^{280}$, the Pre-Industrial. For a comparison of PlioMIP1 and PlioMIP2, see (b) and (e). For the individual effects of $CO_2$ and Pliocene boundary conditions, see (a) and (c), respectively.


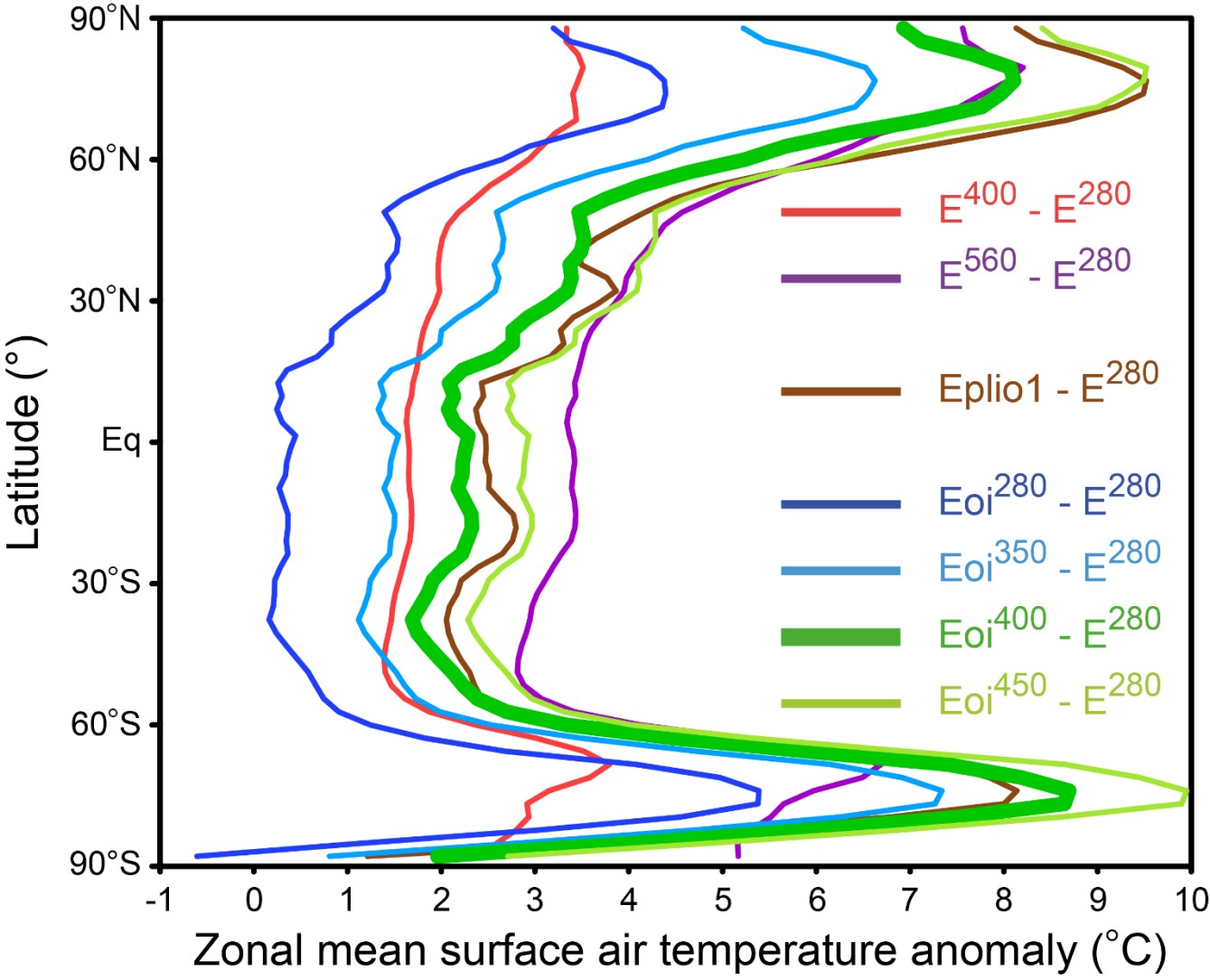

**Figure 4:** **Zonal mean surface air temperature anomaly. The core experiment, Eoi[400], is shown in bold green.**

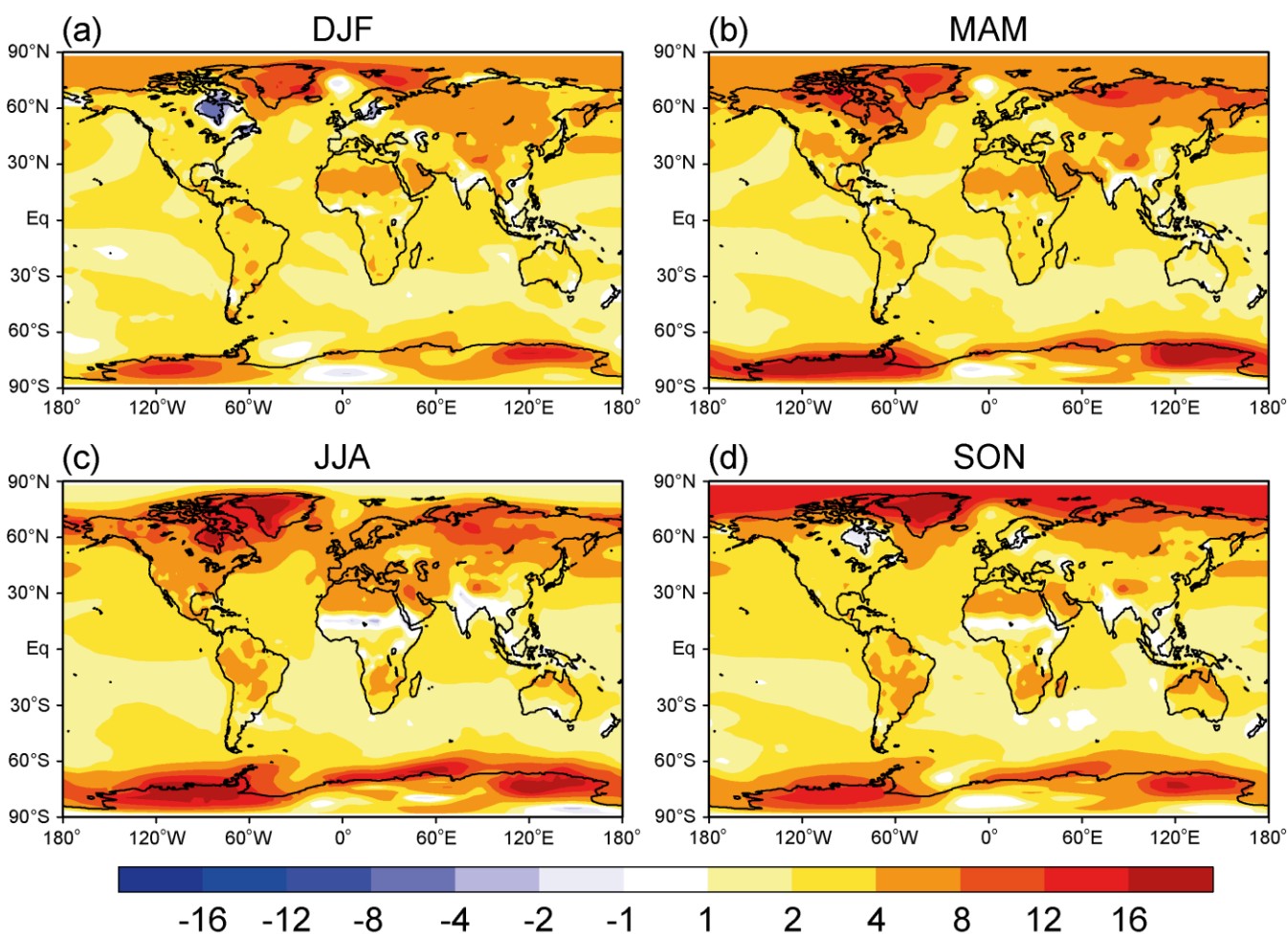

**Figure 5:** **Seasonal surface air temperature anomaly between Eoi[400] and E[280] for December-February (DJF), March-May (MAM), June-August (JJA) and September-November (SON).**

# Annual mean sea surface temperature anomaly (°C)

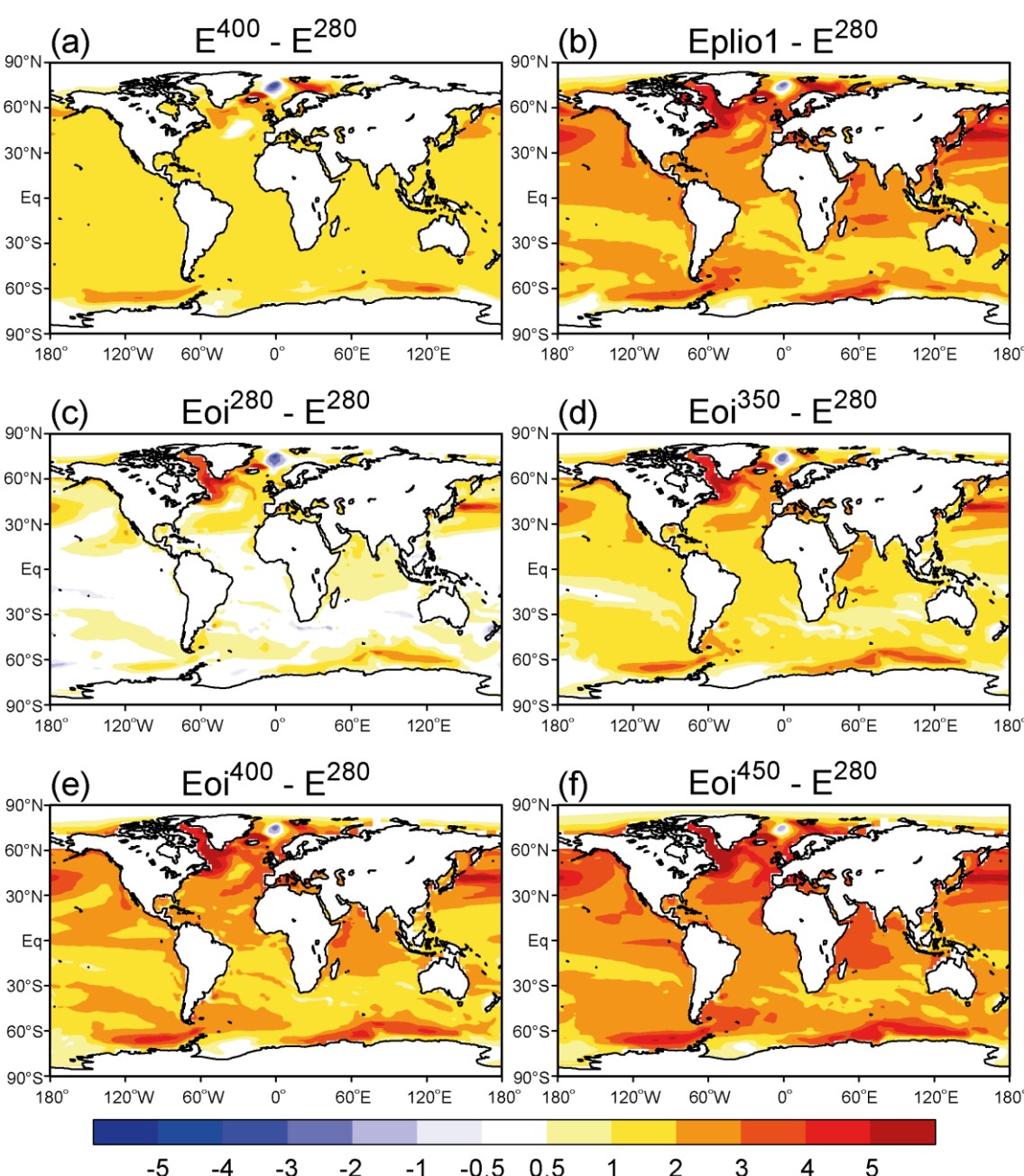

**Figure 6:** Annual mean sea surface temperature anomaly between six experiments and $E^{280}$, the Pre-Industrial. For a comparison of PlioMIP1 and PlioMIP2, see (b) and (e). For the individual effects of $CO_2$ and Pliocene boundary conditions, see (a) and (c), respectively.

# Annual mean precipitation anomaly (%)

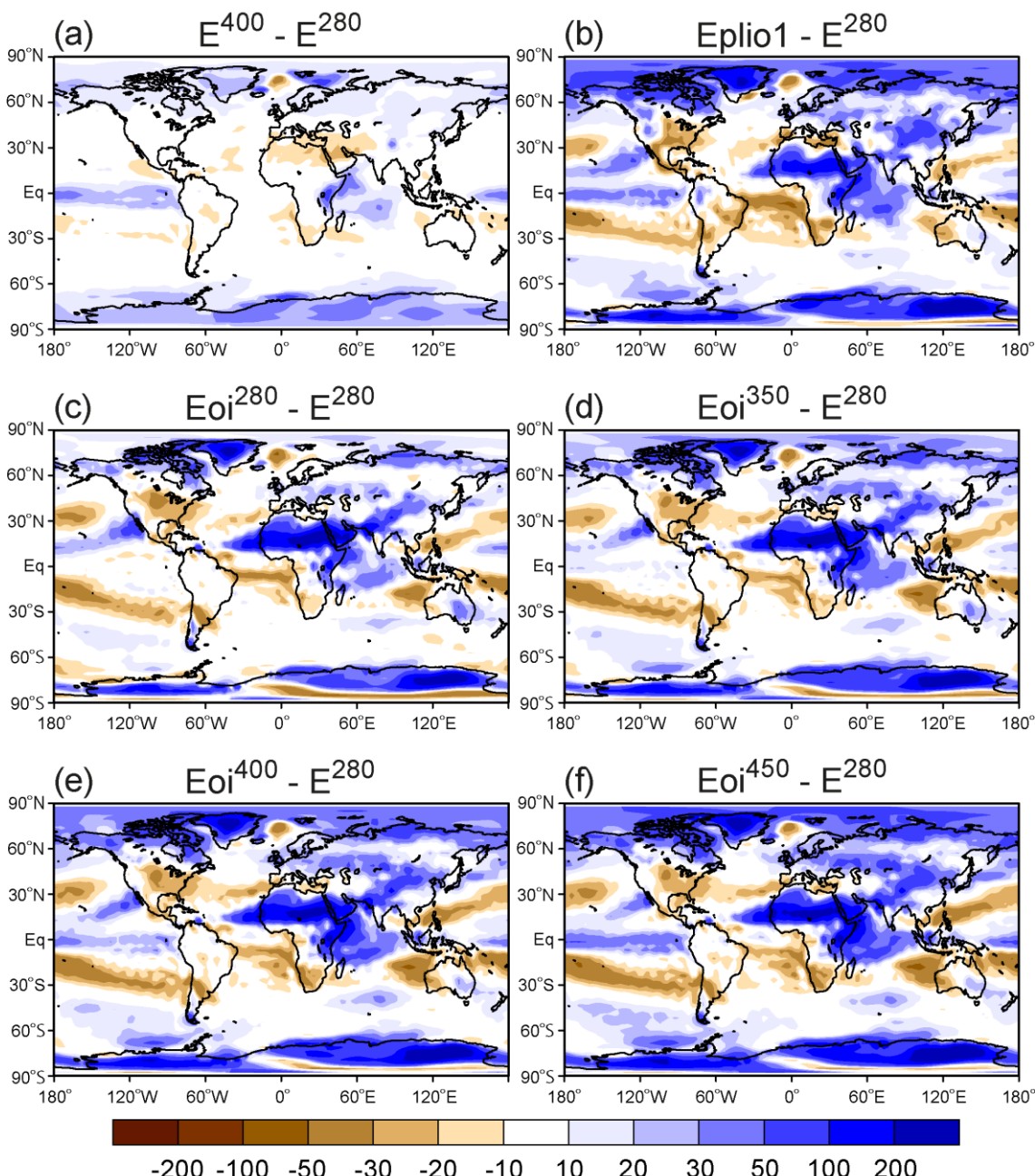

**Figure 7:** **Annual mean precipitation anomaly between six experiments and $E^{280}$, the Pre-Industrial, as a percentage of $E^{280}$, eg.**
**$100 \times (Eoi^{400} - E^{280}) / E^{280}$.** **For a comparison of PlioMIP1 and PlioMIP2, see (b) and (e).** **For the individual effects of $CO_2$ and**
**Pliocene boundary conditions, see (a) and (c), respectively.**

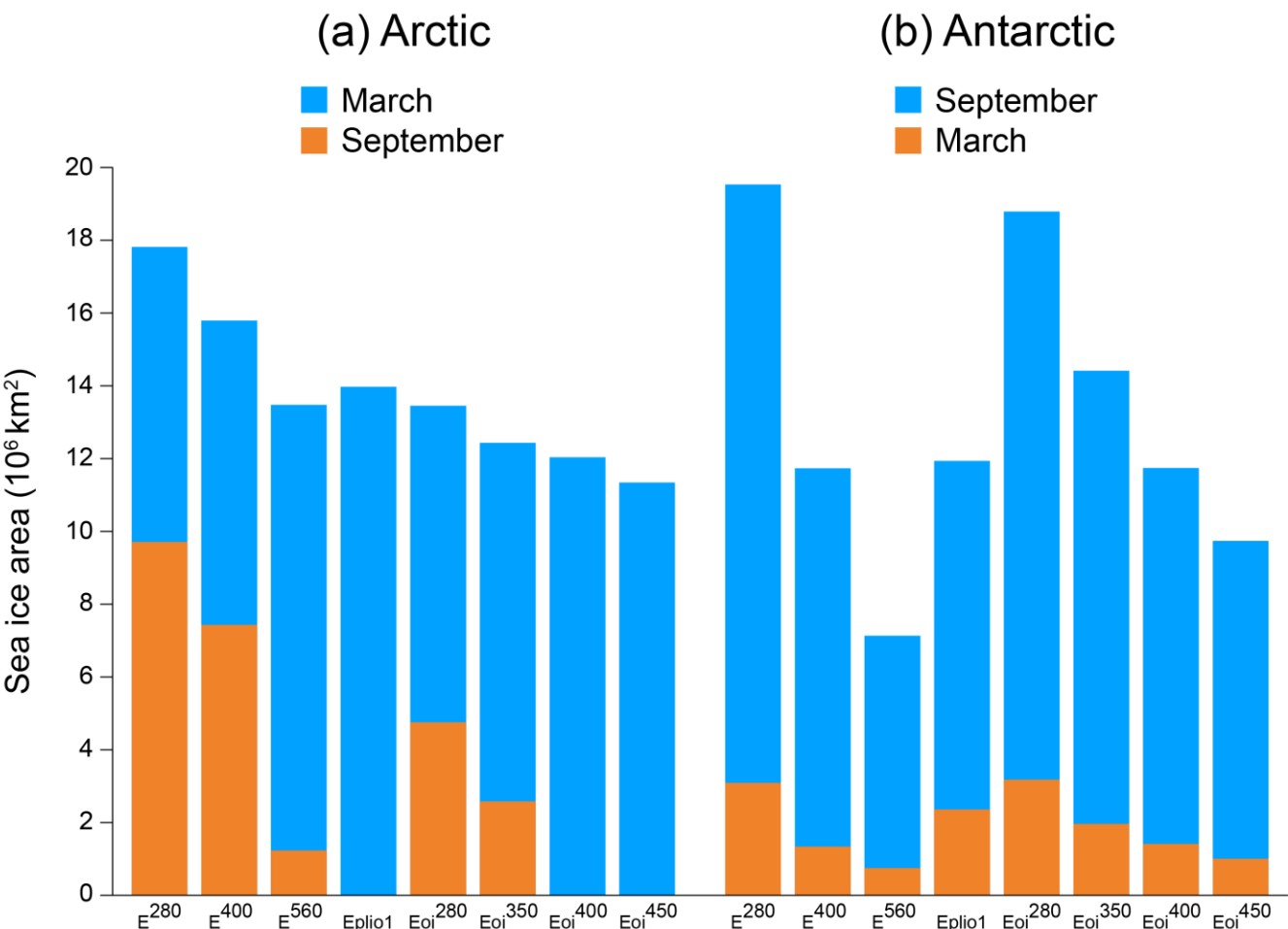

**Figure 8:  Total surface area of sea ice in each polar region during March and September.**

## Mixed layer depth (m) and 15% sea ice conc. during March

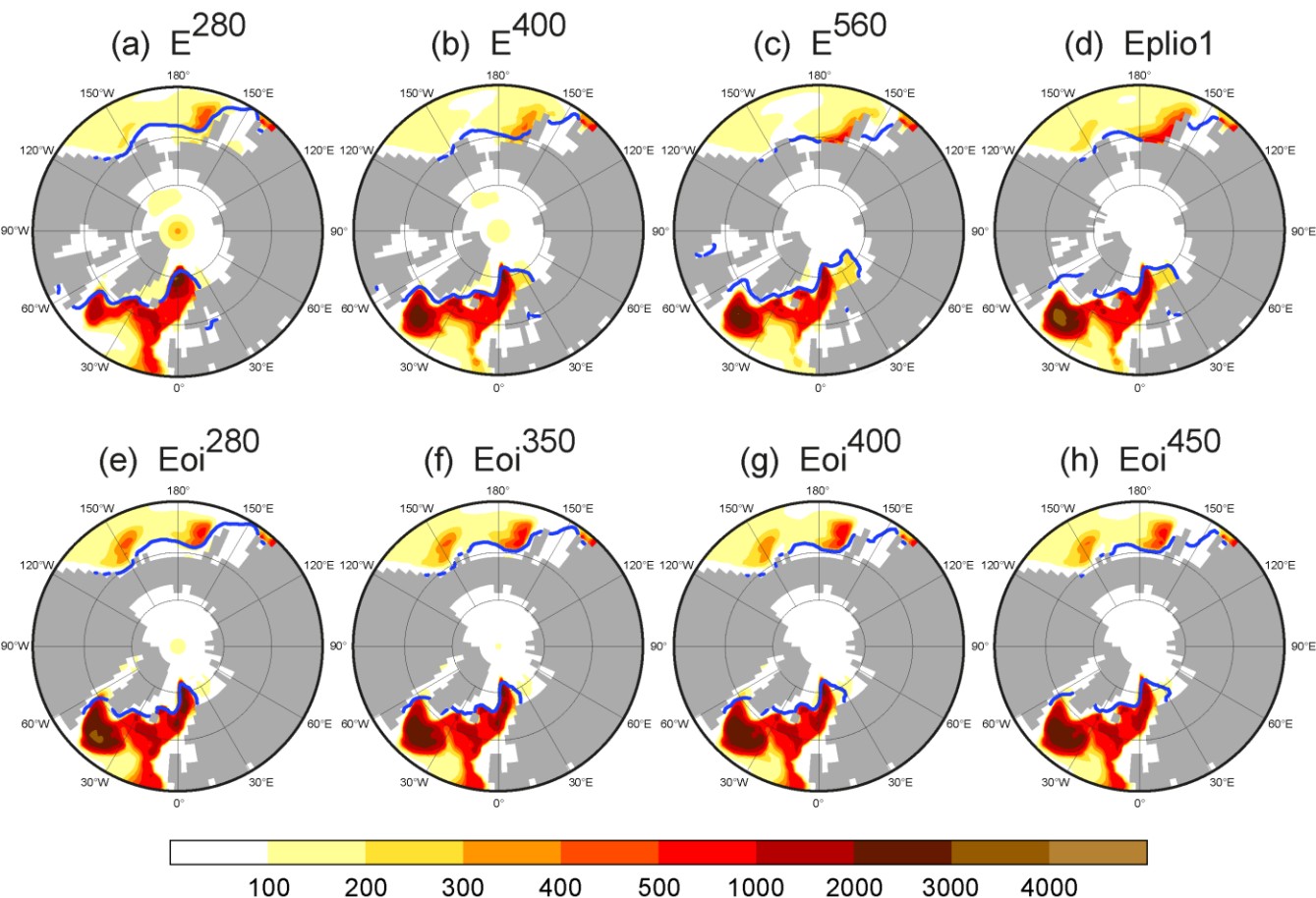

**Figure 9:** Oceanic mixed layer depth in the northern hemisphere during March. The blue lines indicate the extent of 15% sea ice concentration. For a comparison of PlioMIP1 and PlioMIP2, see (d) and (g). For the individual effects of $CO_2$ and Pliocene boundary conditions, see (b) and (e), respectively.

# Mixed layer depth (m) and 15% sea ice conc. during September

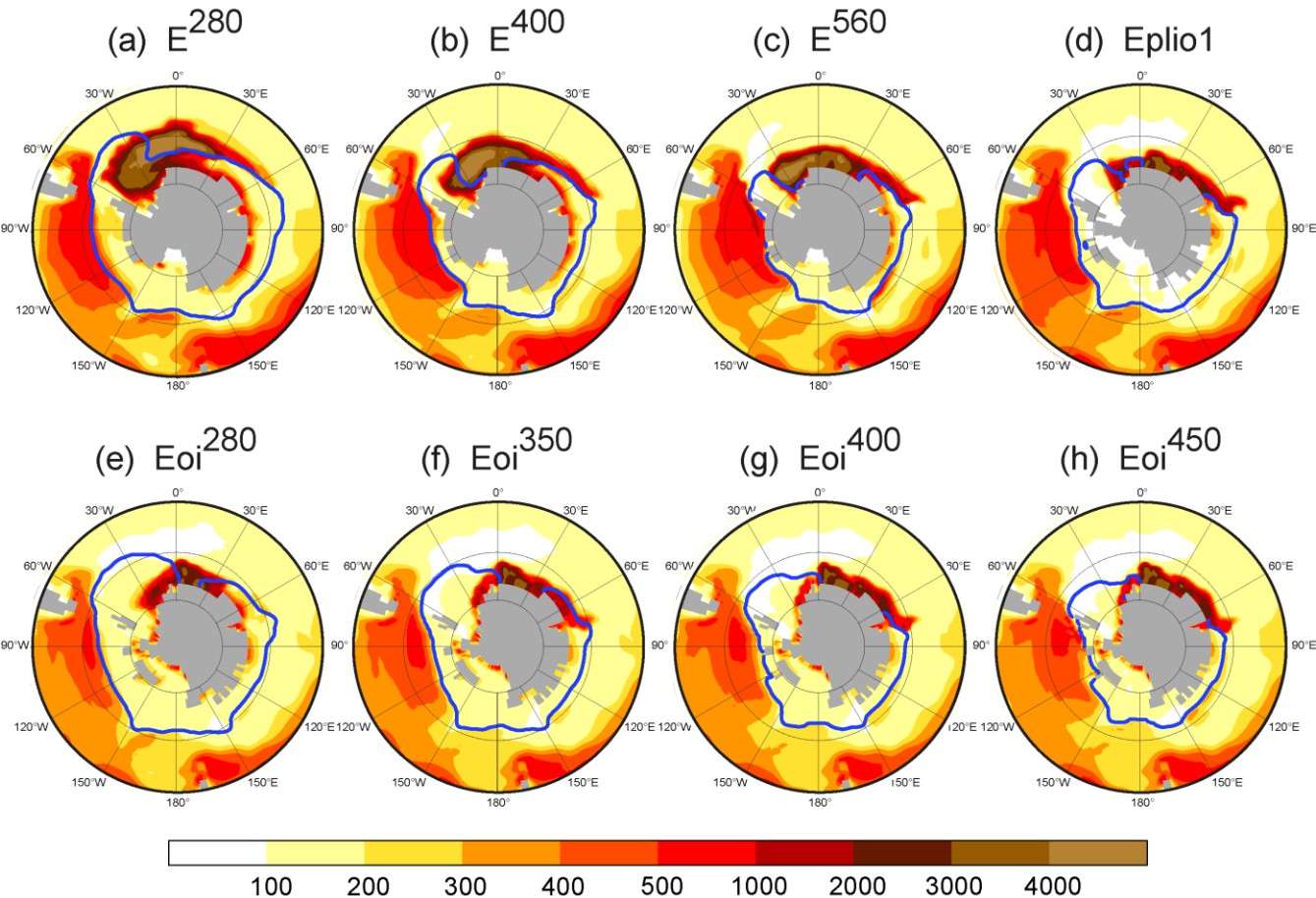

**Figure 10:** Oceanic mixed layer depth in the southern hemisphere during September. The blue lines indicate the extent of 15% sea ice concentration. For a comparison of PlioMIP1 and PlioMIP2, see (d) and (g). For the individual effects of $CO_2$ and Pliocene boundary conditions, see (b) and (e), respectively.

# Atlantic meridional overturning circulation (Sv)

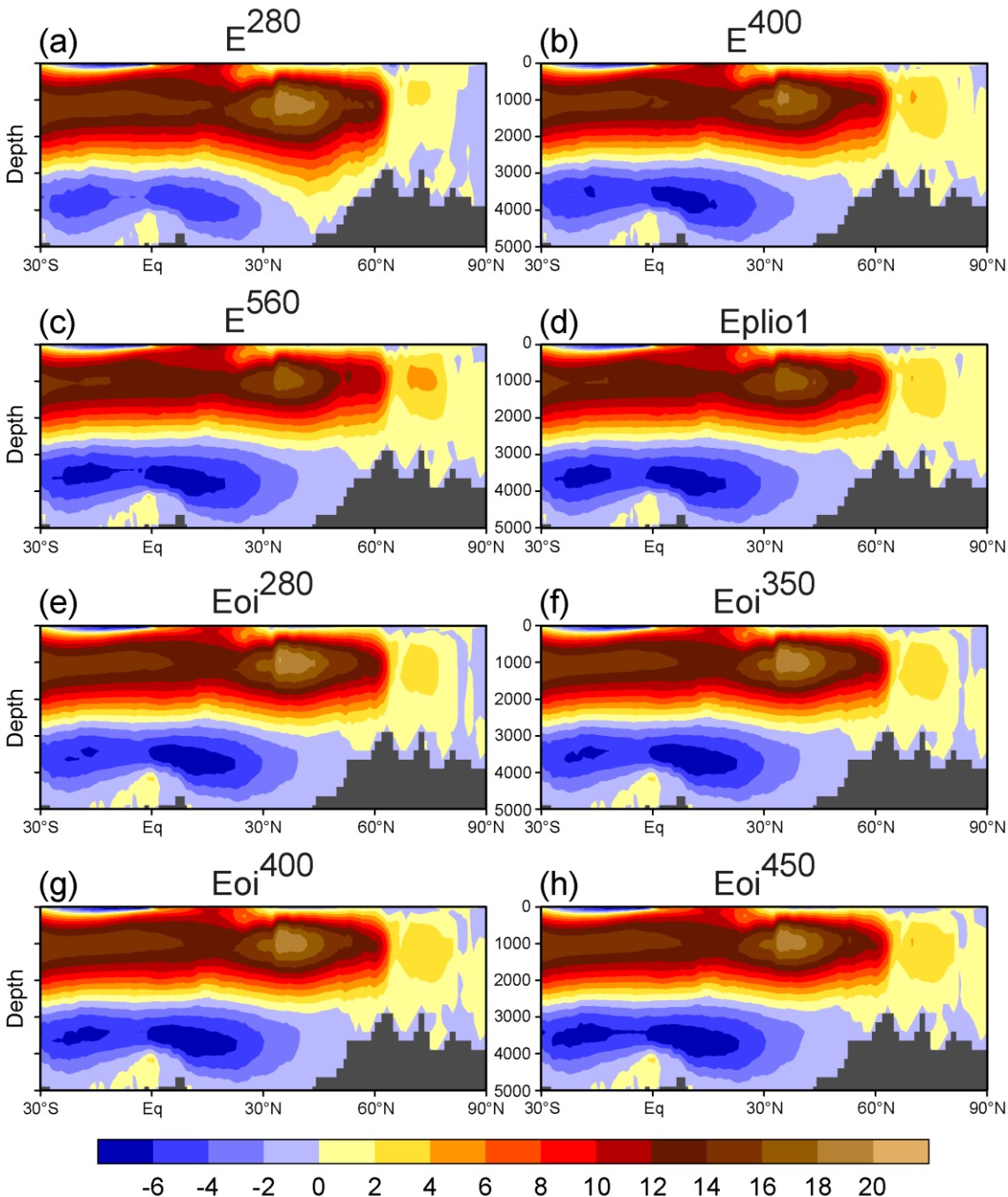

**Figure 11:** **Streamfunction of the Atlantic meridional overturning circulation, averaged over the last 500 years. For a comparison of PlioMIP1 and PlioMIP2, see (d) and (g). For the individual effects of $CO_2$ and Pliocene boundary conditions, see (b) and (e), respectively.**

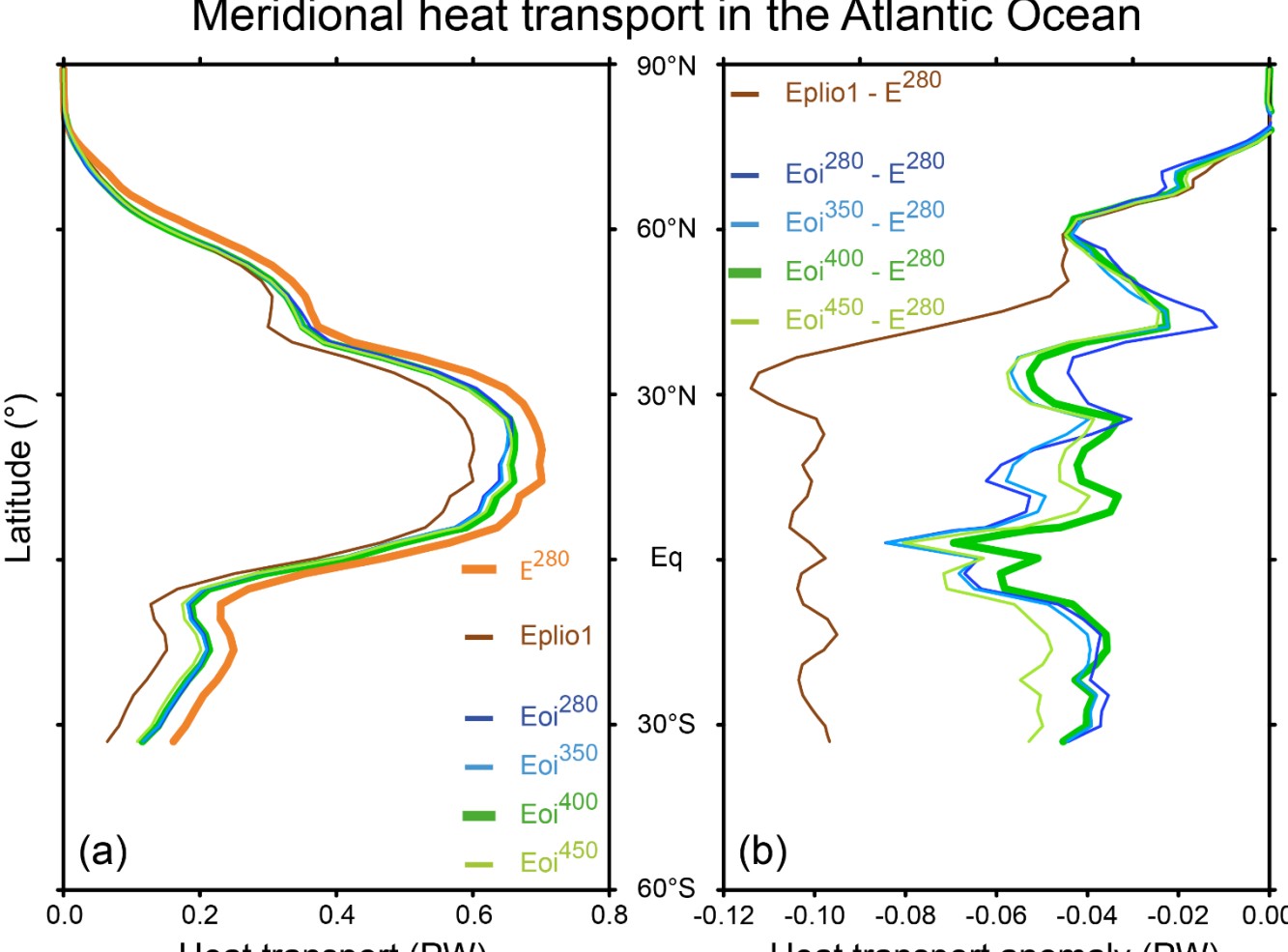

**Figure 12: (a)** The Atlantic Ocean meridional heat transport in the Pliocene experiments and in $E^{280}$ and **(b)** the anomaly between the Pliocene experiments and $E^{280}$. The core experiment, $Eoi^{400}$, is shown in bold green.

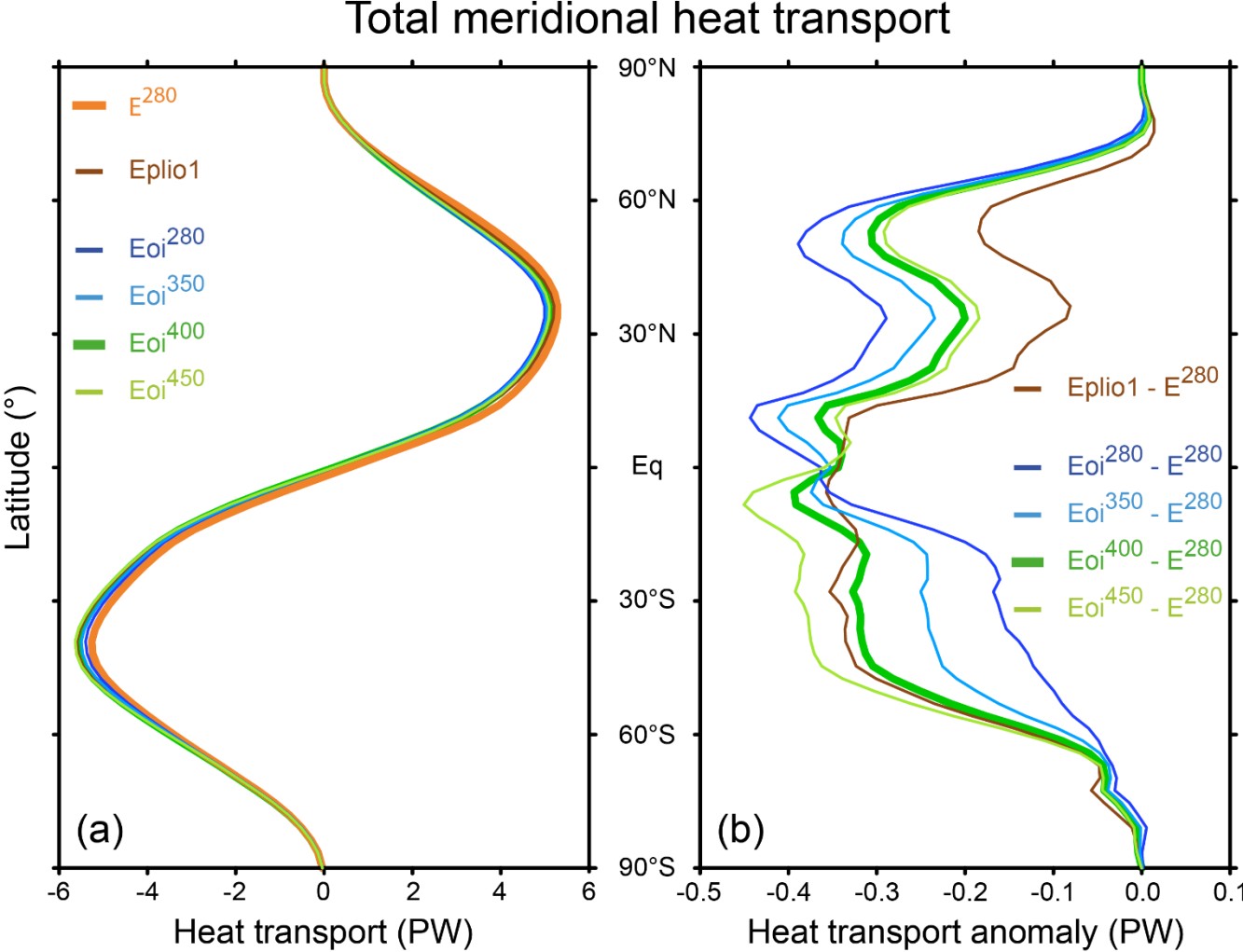

**Figure 13: (a)** The total meridional heat transport in the Pliocene experiments and in $E^{280}$, as calculated from the top-of-atmosphere radiative transport, and **(b)** the anomaly between the Pliocene experiments and $E^{280}$. The core experiment, $Eoi^{400}$, is shown in bold green.

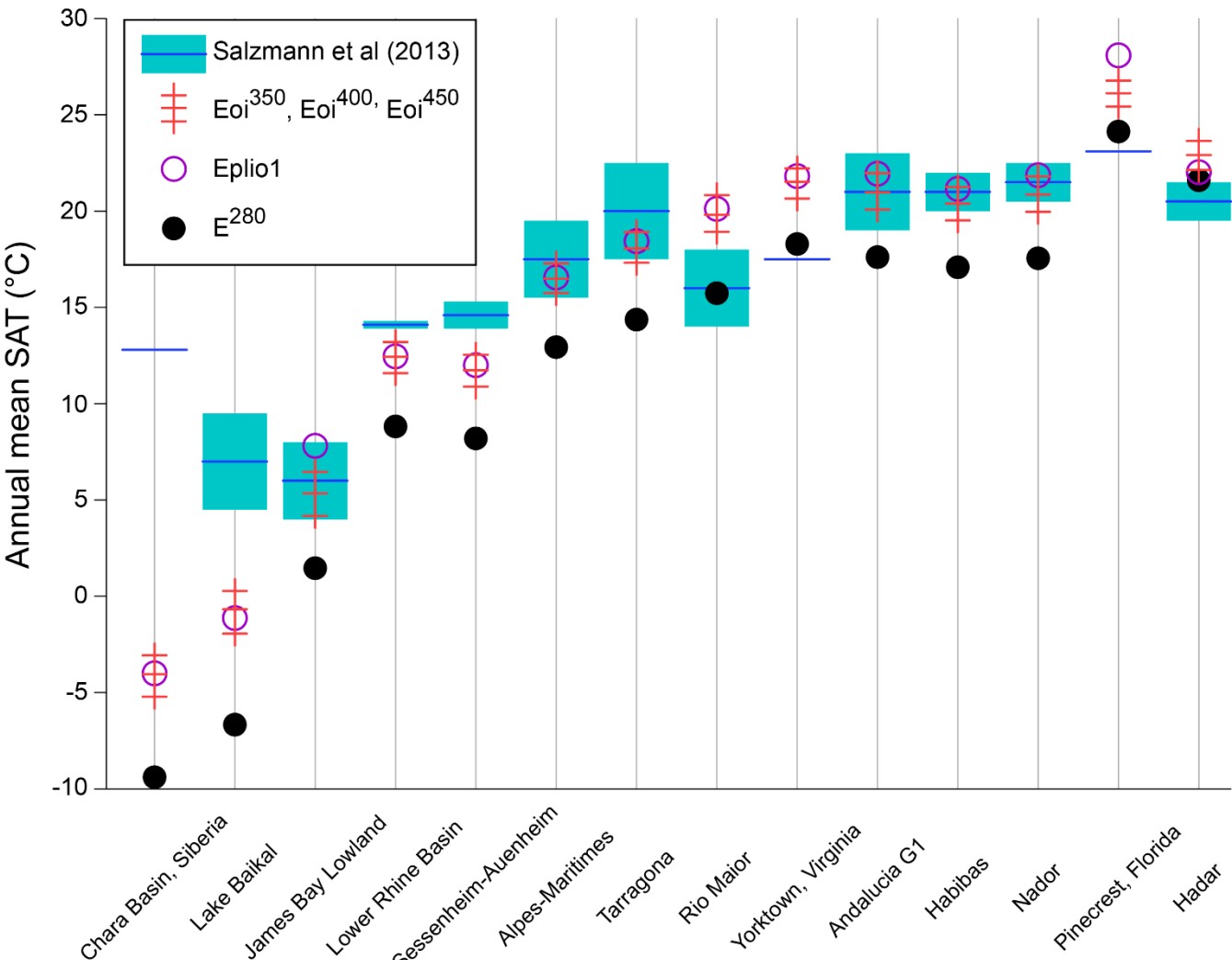

**Figure 14: Comparison of annual mean surface air temperatures derived from marine proxy data (Table S3b of Salzmann et al., 2013) and those from model experiments. Proxy data are indicated by the dark blue line and the uncertainty range by light blue. The three red marks represent, from top to bottom, E450, E400 and E350.**

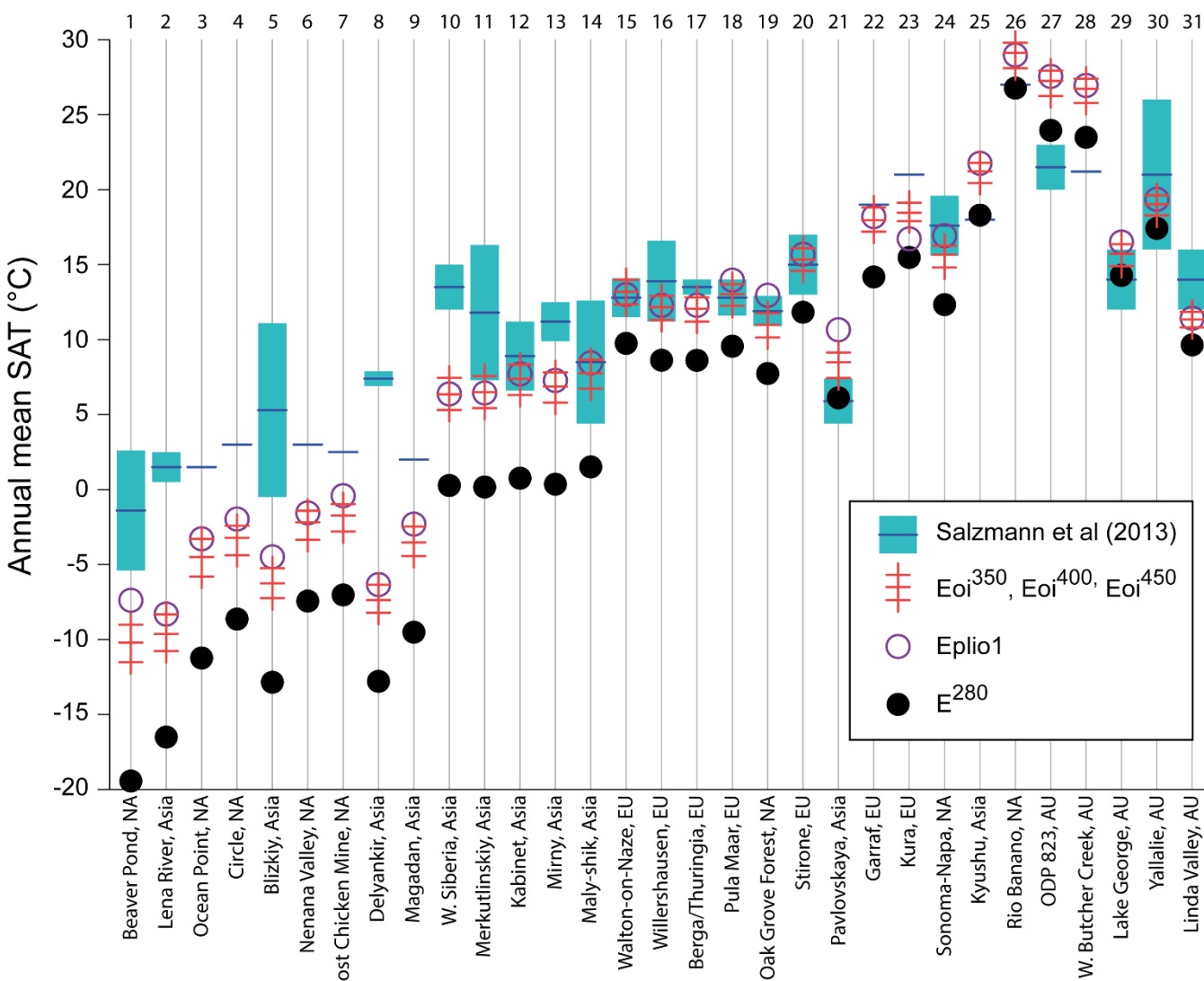

**Figure 15: Comparison of annual mean surface air temperatures derived from terrestrial proxy data (Table S3a of Salzmann et al., 2013) and those from model experiments. Proxy data are indicated by the dark blue line and the uncertainty range by light blue. The three red marks represent, from top to bottom, E[450], E[400] and E[350]. Abbreviations used at the bottom of the figure: NA (North America), EU (Europe), AU (Australia).**


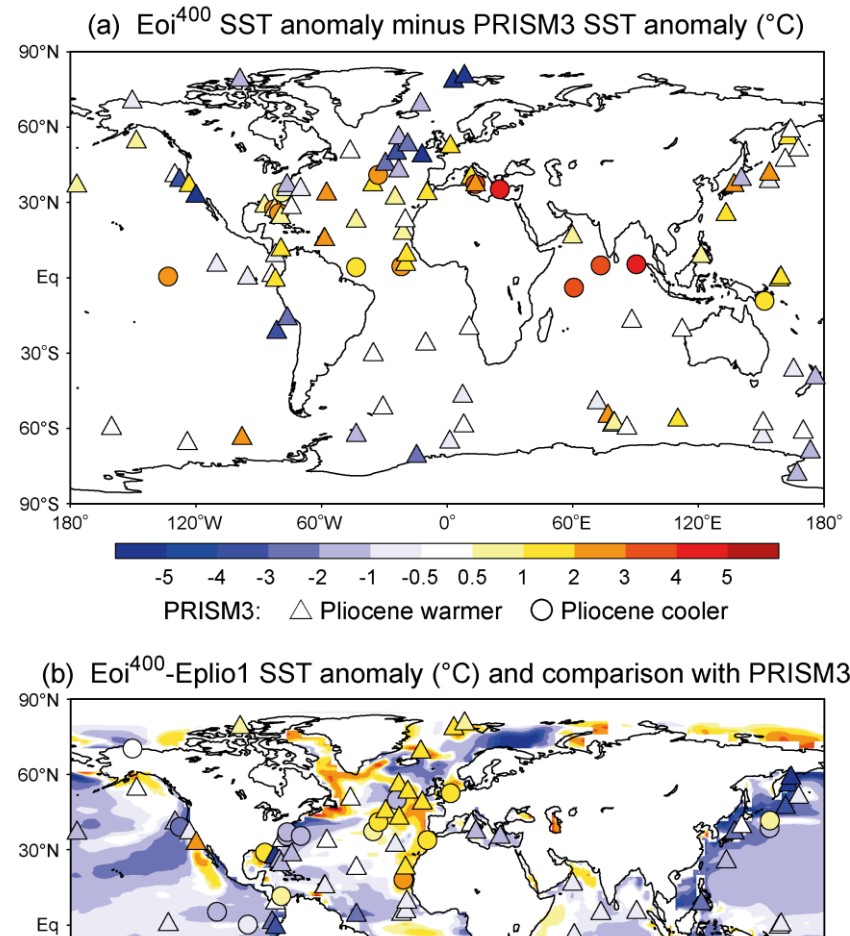

**Figure 16: (a)** Comparison of annual mean model SST anomalies and PRISM3 proxy data SST anomalies. Blue (red) indicates that model SST anomalies are smaller (greater) than those of proxy data. The shape of the symbols indicates whether proxy data suggests higher (triangle) SST in the Pliocene or lower (circle). **(b)** The difference between PlioMIP1 and PlioMIP2 SST. The shape of the symbols at PRISM3 locations indicates whether PlioMIP2 agrees better with proxy data (triangle) or whether PlioMIP1 agrees better (circle).


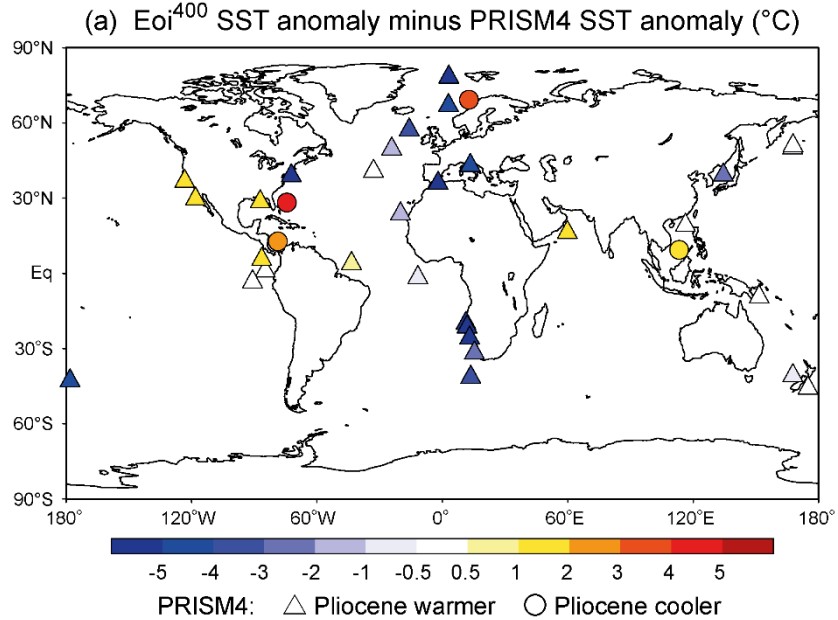

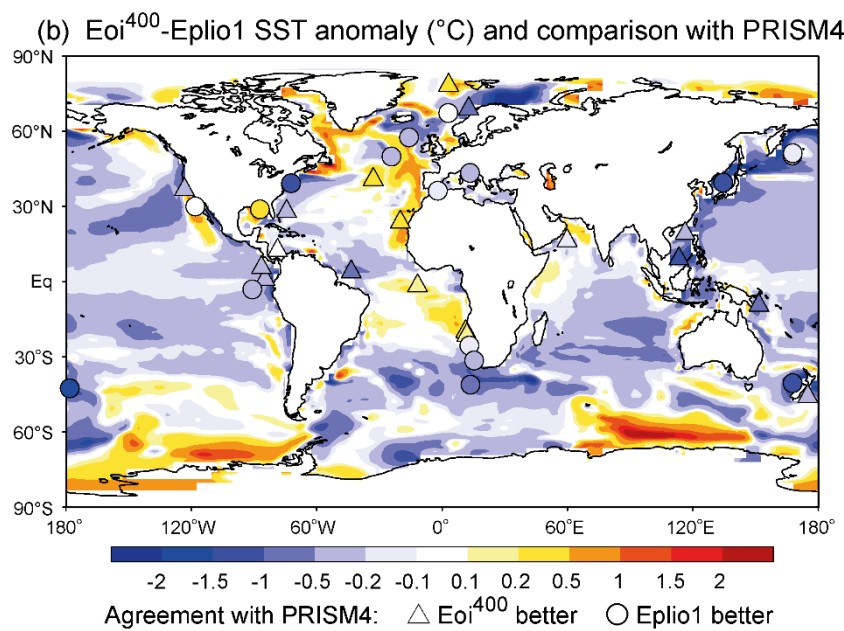

**Figure 17: (a) Comparison of annual mean model SST anomalies and PRISM4 proxy data SST anomalies. Blue (red) indicates that model SST anomalies are smaller (greater) than those of proxy data. The shape of the symbols indicates whether proxy data suggests higher (triangle) SST in the Pliocene or lower (circle). (b) The difference between PlioMIP1 and PlioMIP2 SST. The shape of the symbols at PRISM4 locations indicates whether PlioMIP2 agrees better with proxy data (triangle) or whether PlioMIP1 agrees better (circle).**

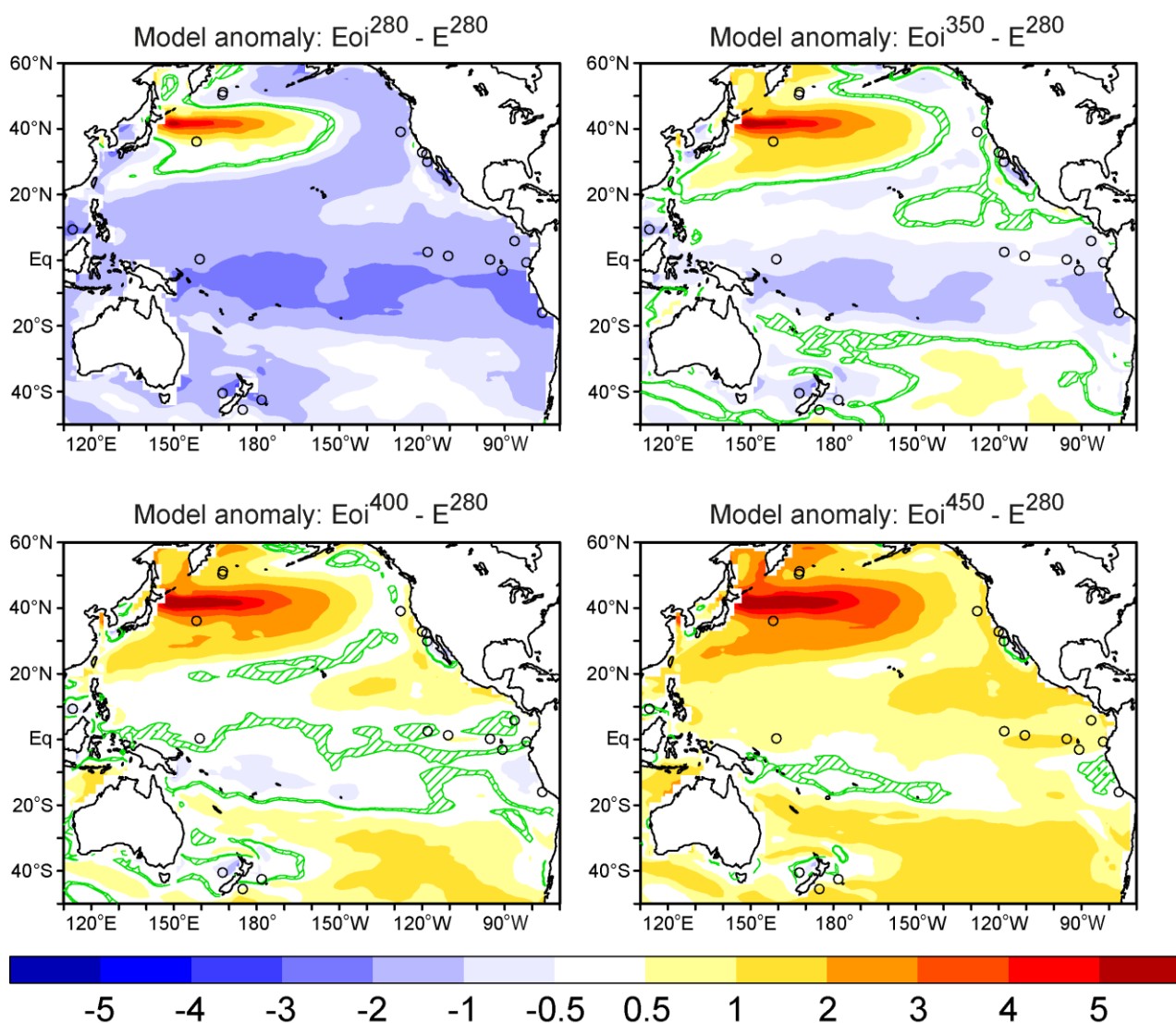

**Figure 18: Comparison of annual mean model SST anomalies and T2019 (Tierney et al, 2019) proxy data SST anomalies. Model anomalies refer to the Pliocene experiments, a) Eoi$^{280}$, b) Eoi$^{350}$, c) Eoi$^{400}$ and d) Eoi$^{450}$. For reference, the location of T2019 core sites are shown as black circles. 95% confidence levels are shown by the green striped area.**

| Expt | Land-sea mask | Topography | Vegetation | Ice sheet | $CO_2$ (ppm) | Integration length (years) |
|------|---------------|------------|------------|-----------|-------------|----------------------------|
| $E^{280}$ | Local modern | Local modern | Local Pre-Ind | Local Pre-Ind | 280 | 2,220 |
| $E^{400}$ | Local modern | Local modern | Local Pre-Ind | Local Pre-Ind | 400 | 3,000 |
| $E^{560}$ | Local modern | Local modern | Local Pre-Ind | Local Pre-Ind | 560 | 3,920 |
| Eplio1 | PRISM3 | Local modern + PRISM3 anom | PRISM3 | PRISM3 | 405 | 4,000 |
| $Eoi^{280}$ | PRISM4 | Local modern + PRISM4 anom | PRISM3 | PRISM4 | 280 | 2,500 |
| $Eoi^{350}$ | PRISM4 | Local modern + PRISM4 anom | PRISM3 | PRISM4 | 350 | 3,000 |
| $Eoi^{400}$ | PRISM4 | Local modern + PRISM4 anom | PRISM3 | PRISM4 | 400 | 4,000 |
| $Eoi^{450}$ | PRISM4 | Local modern + PRISM4 anom | PRISM3 | PRISM4 | 450 | 3,000 |

**Table 1: Boundary conditions, $CO_2$ levels and model integration length for each experiment in the present study.**

| | |
|------|------|
| $CH_4$ | 760ppb |
| $N_2O$ | 270ppb |
| $O_3$ | Local modern |
| Solar constant | 1365 W/m$^2$ |
| Eccentricity | 0.016724 |
| Obliquity | 23.446° |
| Perihelion | 102.04° |

**Table 2: Settings common to all experiments in the present study. These refer to the last 1000 years, before which the following settings were used: 863.303ppb $CH_4$, 279.266ppb $N_2O$, 1366.12 W/m$^2$ solar constant.**

| Expt | Global SAT (°C) | Global ΔSAT (°C) | North SAT (°C) | Tropical SAT (°C) | South SAT (°C) | TOA En Bal (W/m$^2$) | Global Precip (mm/dy) | Global SST (°C) | Global ocean T (°C) | AMOC Index (Sv) |
|---|---|---|---|---|---|---|---|---|---|---|
| E$^{280}$ | 12.8 | 0.0 | -11.4 | 23.0 | -18.4 | 0.88 | 2.69 | 17.0 | 1.91 | 19.5 |
| E$^{400}$ | 14.8 | 2.0 | -8.1 | 24.7 | -15.3 | 0.96 | 2.79 | 18.4 | 2.54 | 18.7 |
| E$^{560}$ | 16.8 | 3.9 | -4.2 | 26.5 | -12.7 | 1.02 | 2.90 | 19.9 | 3.40 | 17.8 |
| Eplio1 | 16.3 | 3.5 | -3.0 | 25.7 | -12.3 | 0.85 | 2.87 | 19.2 | 3.44 | 17.8 |
| Eoi$^{280}$ | 13.9 | 1.1 | -7.7 | 23.5 | -15.0 | 0.80 | 2.74 | 17.5 | 2.12 | 20.2 |
| Eoi$^{350}$ | 15.1 | 2.3 | -5.7 | 24.6 | -12.2 | 0.83 | 2.81 | 18.4 | 2.69 | 20.0 |
| Eoi$^{400}$ | 15.9 | 3.1 | -4.4 | 25.4 | -12.0 | 0.84 | 2.86 | 19.0 | 3.09 | 20.0 |
| Eoi$^{450}$ | 16.6 | 3.8 | -3.2 | 26.0 | -11.0 | 0.84 | 2.90 | 19.6 | 3.46 | 19.8 |

800

**Table 3: Global mean and other values for each experiment, averaged over the last 100 years. 1) Experiment name, 2) Global surface air temperature, 3) Global surface air temperature anomaly with E$^{280}$ as the reference, 4) Surface air temperature averaged over latitudes 60°N-90°N, 5) Surface air temperature averaged over latitudes 30°S-30°N, 6) Surface air temperature averaged over latitudes 90°S-60°S, 7) Energy balance at the top of the atmosphere, 8) Global precipitation, 9) Global sea surface temperature, 10) Global ocean temperature averaged across all depths, 11) AMOC index. The AMOC index is averaged over the last 500 years to remove large centennial variability, as seen in Figure 2.**

| Experiment | Global | 90°N-60°N | 30°N-30°S | 60°S-90°S |
|---|---|---|---|---|
| Eplio1 | 1.05 | -1.28 | 2.19 | 0.04 |
| Eoi$^{280}$ | -0.76 | -3.03 | 0.23 | -1.08 |
| Eoi$^{350}$ | 0.14 | -4.43 | 1.21 | -0.35 |
| Eoi$^{400}$ | 0.76 | -1.92 | 1.88 | 0.20 |
| Eoi$^{450}$ | 1.31 | -1.40 | 2.43 | 0.74 |

810

**Table 4: The difference between the annual mean SST anomaly (from E$^{280}$) of each experiment and the equivalent from the PRISM3 proxy data. Positive values signify that the increase in model SST is greater than the increase in proxy SST. Units are °C. The PRISM3 proxy data were used as SST boundary conditions for AGCM experiments in PlioMIP1.**