# Peer review of "PlioMIP2 simulations using the MIROC4m climate model"

_Climate of the Past, 2020_

## Referee Comment (RC1) · Anonymous Referee #1 · 15 Mar 2020

I enjoyed reading the manuscript "PlioMIP2 simulations using the MIROC4m climate model". It is well-written. The figures are well-made. The results and analysis are clear and logical. I am in support of publishing this study. Here are a few thoughts that the authors might consider:

1. I am afraid, the conclusion (also in the abstract) that $CO_2$ below 400 ppm provides a better simulation, could be better supported. This conclusion is derived from comparing the global mean SST difference between simulated SST and PRISM3 SSTs, and "Since the global difference is determined more by the low latitudes, Eoi350 gives the best global fit (a difference of 0.14°C)" (citing the original text). However, it is worth noting that 1) tropical SST estimates have uncertainties (Tierney et al., 2019, GRL) and 2) Eoi400 does provide better match with mid- and high latitudes' SSTs. The proxy

data-model comparison can benefit from Student-t test by considering the proxy data uncertainty and spatial spread as well as the model data spatial spread.

2. Further, PlioMIP2 is designed to focus on KM5C in order to avoid ambiguities of model-proxy data comparison introduced by orbital cycles (Haywood et al., 2016). As part of this effort, datasets from PRISM4 (cited in the manuscript) and PlioVAR (Mc-Clymont et al., 2020) are developed. PRISM3 dataset included more tropical sites than PRISM4 or PlioVAR. One might wonder that if the same analysis discussed in section 4.8 were done with PRISM4 or PlioVAR dataset, the conclusion might be different.

3. Section 4.1: the description of simulated surface air temperature has mentioned a few times that the CO2 induced warming is homogenous (uniform). However, by examining Fig. 3 and 4, it looks that there is a clear polar amplification signal with increasing CO2. It shows up in Fig. 3a and when comparing Fig. 4 Eoi experiments with different CO2.

Other comments: 1. line 9: please spell out PMIP.

2. Line 23: PlioMIP2 is designed to simulated KM5C, this sentence hints that the current simulation is not using realistic orbit, which is not the case.

3. Line 18, please also reference more recent publications, here are a few examples: For pollen:

Panitz, S., Salzmann, U., Risebrobakken, B., De Schepper, S. and Pound, M.J., 2015. Climate variability and long-term expansion of peat lands in Arctic Norway during the late Pliocene (ODP Site 642, Norwegian Sea). Climate of the Past, 11, pp.5755-5798. For Fossil floras: Fletcher, T., Feng, R., Telka, A.M., Matthews Jr, J.V. and Ballantyne, A., 2017. Floral dissimilarity and the influence of climate in the Pliocene High Arctic: Biotic and abiotic influences on five sites on the Canadian Arctic Archipelago. Frontiers in Ecology and Evolution, 5, p.19.

4. Line 65 – 68: Here is another important reference for CO2 using boron isotope:

Martínez-Botí, M.A., Foster, G.L., Chalk, T.B., Rohling, E.J., Sexton, P.F., Lunt, D.J., Pancost, R.D., Badger, M.P.S. and Schmidt, D.N., 2015. Plio-Pleistocene climate sensitivity evaluated using high-resolution CO 2 records. Nature,

5. Line 85 – 86, I don't understand this. Is the flux change still happening between atmosphere and ocean?

6. Line 90, what is the sigma level referred to? P/Ps?

7. Line 93 – 95, What is the cloud scheme? Is there microphysics? Does the model simulate indirect effects of aerosols? or the direct effect? or none?

8. Line 98, "Water and heat exchange between the land surface and atmosphere, and runoff flux to a river routing model are represented." What does "represented" mean? Is there a multi-layer radiation scheme or simple scaling?

9. Line 104, does the sigma level here mean potential density level? Please specify.

10. Line 111, what about the radiation scheme for sea ice? Is it simple scaling with fixed end members of albedo? or more sophisticated radiation scheme?

11. Line 136, why did the doubling CO2 experiment use 571 ppm instead of 570 (285x2)?

12. Line 46 – 47, Please specify that paleogeography is also changed according to the PlioMIP2 protocol.

13 line 160, This whole sentence is very confusing. I don't think it is needed. Please remove.

14 Line 194 – 195, I don't understand what do you mean by "bias"?

15 Line 254 – 262, it would be help to use present-day known deep water formation sites (such as GIN sea, Lab sea) as the geographic reference to construct this section (e.g. deep water formation weakens in the GIN sea, but strengthens in Lab sea).

16 Line 286, the authors might consider including another recent simulation in this list: Feng, R., Bette L, O.B., Brady, E.C. and Rosenbloom, N.A., 2020. Increasing Earth System Sensitivity in mid-Pliocene simulations from CCSM4 to CESM2. https:/doi.org/10.1002/essoar.10501546.1

17 Line 304 to 305: It might worth noting that this is different from the recent PlioMIP2 runs with CCSM4/CESM1 and 2 (the study listed in the above) (heat transport are reduced in both hemispheres).

18 Line 350 – 351, "It is worth noting that not only does Eoi280 give the best fit at low latitudes, but bucks the trend at the northern high latitudes where the discrepancy is smaller than that for Eoi350." I don't understand this sentence.

19 Line 390-391, "with modelled climate, ice sheets and vegetation exhibiting strong regional variations associated with orbital parameters, whether as time-dependent forcing in transient simulations (Willeit et al., 2013) or fixed to minimum or maximum forcings (Dolan et al., 2011)." Feng et al., (2017, EPSL) further demonstrated the regional warming/cooling patterns induced by mPWP orbital variations in a statistically meaningfully way.

---

## Referee Comment (RC2) · Anonymous Referee #2 · 24 Mar 2020

This manuscript presents initial results from the MIROC4m modelling group for 8 experiments according to the PlioMIP 2 design. All these simulations have been integrated for long term. The analysis work is sufficient and logical, some previous PlioMIP2 studies have been discussed with their outputs. This is a solid contribution to the PlioMIP2 effort. I recommend publication of this paper with considering the minor suggestions listed as below:

General comments: 1. The authors have included the EPlio1 experiment in the present study to compare with the PlioMIP2 core experiment. However, it lacks broadly discussion about the difference between these two experiments with regards to the SAT, SST, precipitation and sea ice. The Eplio1 differs from the PlioMIP2 not only in the topography, which is moslty mentioned in the results, but also in the land-sea mask, the lakes

[Figure]

in the Africa as well as the soil types. I suggest the authors could pay more attention with regards to these aspects.

2. Section 4.6: The Meridional heat transport have been studied in many PlioMIP 2 papers (Li et al.,2020, Chandan and Peltier et al.,2017, Tan et al.,2020, Feng et al.,2020 etc). I suggest the authors can compare the results with the others. According to Figure 13, it is worth noting that the import change appears over the low latitudes which is different from the aforementioned studies. By the way, Figure 13 is not easy to read, I suggest to put all the absolute values in one plot.

3. Section 4.7: I do not understand why the authors compare the E280 with the proxy data reconstructed for the mid-Pliocene and discuss the performance of the data-model fit of the Pliocene experiment and the Pliocene experiment together (line 316). Section 4.8: I suggest the authors could also compare the model outputs with the PRIMS4 data (Foley and Dowsett, 2019) which is more specific for the PlioMIP phase 2.

Specific comments:

1. Line 145: Confused. Please specify which 4 experiments are further integrated for another 1000 years and why they need to be integrated for another 1000 years . 2. Line 155: Why are the greenhouse gases not changed to be PlioMIP level in the first 3000 years run ? 3. Line 167: In my opinion, figure 3a does not approve "Temperature increase fairly uniformly", In figure 3a, the warming amplitude over high latitudes is stronger than the low latitudes, the land warming is larger than the ocean broadly, and some extreme warming regions are found in the Barents sea and around coastal regions of the Antarctic. 4. Line 184: Compared to the PlioMIP1, PlioMIP2 also modify the Bering strait and Northern Canadian Archipelago regions, do they have any impacts on this different warming ? 5. Line 196:"near-uniform increase outside the polar regions" is not appropriate, according to Figure 4, I suggest to be "near-uniform increase outside the low latitudes (30N-30S)". 6. Line 194: What is the "bias" here ? 7. Line 205: What is the reason for the seasonal change of the artic region , is the sea ice

fraction change responsible for that ? 8. Line 221: What is the potential reason for the SST anomalies for the other regions , e.g. the Labrador sea, the North Atlantic Ocean ?

---

## Author Comment (AC1) · 9 May 2020

We are grateful to the reviewer for his/her comments which have helped us to identify deficiencies in our manuscript. Below, we reply to each of the comments which are in red font.

1. I am afraid, the conclusion (also in the abstract) that CO2 below 400 ppm provides a better simulation, could be better supported. This conclusion is derived from comparing the global mean SST difference between simulated SST and PRISM3 SSTs, and "Since the global difference is determined more by the low latitudes, Eoi350 gives the best global fit (a difference of 0.14°C)" (citing the original text). However, it is worth noting that 1) tropical SST estimates have uncertainties (Tierney et al., 2019, GRL) and 2) Eoi400 does provide better match with mid- and high latitudes' SSTs. The proxy data-model comparison can benefit from Student-t test by considering the proxy data uncertainty and spatial spread as well as the model data spatial spread.

The conclusion concerning the multi-$CO_2$ experiments was applicable to comparisons with PRISM3 data only. In hindsight, the difficulties in making a proper and thorough assessment should have been emphasized – not only because of the various regional discrepancies but also, as the reviewer has pointed out, uncertainties related to the proxy data themselves. We have referred to Tierney et al (2019), compared the SST anomalies with those of the multi-$CO_2$ experiments, and following the reviewer's suggestion, performed a Student's t-test. In the tropical Pacific, Eoi$^{400}$ SST agrees well with the data from Tierney et al (2019). This highlights the continuing need for climate reconstructions from proxy data. We have made some modifications in the abstract and conclusion to reflect our latest results.

2. Further, PlioMIP2 is designed to focus on KM5C in order to avoid ambiguities of model-proxy data comparison introduced by orbital cycles (Haywood et al., 2016). As part of this effort, datasets from PRISM4 (cited in the manuscript) and PlioVAR (Mc-Clymont et al., 2020) are developed. PRISM3 dataset included more tropical sites than PRISM4 or PlioVAR. One might wonder that if the same analysis discussed in section 4.8 were done with PRISM4 or PlioVAR dataset, the conclusion might be different.

A comparison with SST data from PRISM4 sites was shown in supplementary figure 2. This figure has now been moved to the main paper to ensure that there is at least one figure comparing model results with the recent PRISM4 data. In the supplement, we now include a figure documenting a similar comparison, but for all of the multi-$CO_2$ experiments. For all $CO_2$ values used in our study, the Pliocene experiments are generally unable to reproduce the degree of warming seen in PRISM4 in the northern North Atlantic and the subtropical South Atlantic. We leave analyses with PlioVAR dataset for the future.

3. Section 4.1: the description of simulated surface air temperature has mentioned a few times that the CO2 induced warming is homogenous (uniform). However, by examining Fig. 3 and 4, it looks that there is a clear polar amplification signal with increasing CO2. It shows up in Fig. 3a and when comparing Fig. 4 Eoi experiments with different CO2.

While the Eoi experiments show much greater regional changes in comparison to the experiments with a simple increase in $CO_2$, we agree with the reviewer that the latter experiments themselves do not show uniform warming. On the whole, there is more warming over the continents, which was already stated in the original text. There is a clear polar amplification signal, albeit smaller than that seen in the Eoi experiments and we have made sure to include this in the revised text.

Other comments

1. line 9: please spell out PMIP.

Done.

2. Line 23: PlioMIP2 is designed to simulated KM5C, this sentence hints that the current simulation is not using realistic orbit, which is not the case.

The sentence has been modified to refer to possible future studies related to the mPWP outside of KM5c.

3. Line 18, please also reference more recent publications, here are a few examples:
For pollen: Panitz, S., Salzmann, U., Risebrobakken, B., De Schepper, S. and Pound, M.J., 2015. Climate variability and long-term expansion of peat lands in Arctic Norway during the late Pliocene (ODP Site 642, Norwegian Sea). Climate of the Past, 11, pp.5755-5798. For Fossil floras: Fletcher, T., Feng, R., Telka, A.M., Matthews Jr, J.V. and Ballantyne, A., 2017. Floral dissimilarity and the influence of climate in the Pliocene High Arctic: Biotic and abiotic influences on five sites on the Canadian Arctic Archipelago. Frontiers in Ecology and Evolution, 5, p.19.

Thank you for giving these examples. We have now referred to these papers in the revised manuscript. The volume and page numbers for the first reference differ to those given above.

4. Line 65 – 68: Here is another important reference for CO2 using boron isotope: Martínez-Botí, M.A., Foster, G.L., Chalk, T.B., Rohling, E.J., Sexton, P.F., Lunt, D.J., Pancost, R.D., Badger, M.P.S. and Schmidt, D.N., 2015. Plio-Pleistocene climate sensitivity evaluated using high-resolution CO2 records, Nature.

Thank you. This has been added to the text.

5. Line 85 – 86, I don't understand this. Is the flux change still happening between atmosphere and ocean?

Yes. The sea ice model also acts as the coupled model's air-sea interface, even when the grid point in question is ice-free, in which case the flux is unaffected in the sea ice model.

6. Line 90, what is the sigma level referred to? P/Ps?

Yes, the sigma levels are pressure levels scaled with the surface pressure. This has been added to the text for clarity.

7. Line 93 – 95, What is the cloud scheme? Is there microphysics? Does the model simulate indirect effects of aerosols? or the direct effect? or none?

A prognostic Arakawa-Schubert cumulus scheme and a prognostic cloud water scheme for large-scale condensation are included in the model. Yes, there is microphysics. Direct effects of aerosols are considered in the radiation scheme, making use of hydroscopic growth and refractive indices of aerosols. Indirect effects of aerosols are considered for condensation in stratus clouds. The reader is advised to refer to K-1 model developers (2004) for further details.

8. Line 98, "Water and heat exchange between the land surface and atmosphere, and runoff flux to a river routing model are represented." What does "represented" mean? Is there a multi-layer radiation scheme or simple scaling?

We have rephrased this sentence. Within the land-surface model (MATSIRO), water and heat exchange between the land surface and atmosphere is computed. Within the same model, runoff on the land is also calculated and passed over to a river routing model which transports the runoff water to the ocean model at river mouths. The radiation scheme is multi-layer.

9. Line 104, does the sigma level here mean potential density level? Please specify.

The sigma level represents a normalised geopotential height and takes the value 1 at the free surface and 0 at a fixed depth above which the sigma coordinate system is applied. We have added this description to the text.

10. Line 111, what about the radiation scheme for sea ice? Is it simple scaling with fixed end members of albedo? or more sophisticated radiation scheme?

Upward longwave radiative flux is calculated according to the Stefan-Boltzmann law with an emissivity of 0.95. The albedo of bare ice surface is set to a constant value of 0.5, and that of snow-covered surface varies between 0.65 and 0.85, depending on the temperature. The air-sea/ice flux is calculated by taking into account downward shortwave and longwave radiative fluxes calculated in the atmospheric model. Penetration of shortwave radiative flux into the snow or sea ice is not taken into account.

11. Line 136, why did the doubling CO2 experiment use 571 ppm instead of 570 (285x2)?

On line 135, it was stated that the original value for the Pre-Industrial was set to approximately 285ppm. The exact value is actually 285.431ppm and is now used in the text. Thus, for doubling $CO_2$, the value is about 571ppm.

12. Line 46 – 47, Please specify that paleogeography is also changed according to the PlioMIP2 protocol.

Done.

13. line 160, This whole sentence is very confusing. I don't think it is needed. Please remove.

With all due respect, I think this sentence is needed since we need to distinguish between experiments with Pliocene boundary conditions (Eplio1 and Eoi$^{xxx}$) and those with simply an increase in $CO_2$ levels (E$^{xxx}$), and we refer to the former group of experiments frequently in the text. We have added "Eplio1 and Eoi$^{xxx}$" to make the sentence clearer.

14. Line 194 – 195, I don't understand what do you mean by "bias"?

We have changed the sentence to indicate a larger anomaly (ie greater warming) instead of using the word "bias".

15. Line 254 – 262, it would be help to use present-day known deep water formation sites (such as GIN sea, Lab sea) as the geographic reference to construct this section (e.g. deep water formation weakens in the GIN sea, but strengthens in Lab sea).

This section has been rewritten to discuss each of the deepwater formation sites separately, in the order: the Labrador Sea, the Norwegian Sea and the region west of the British Isles.

16. Line 286, the authors might consider including another recent simulation in this list: Feng, R., Bette L, O.B., Brady, E.C. and Rosenbloom, N.A., 2020. Increasing Earth System Sensitivity in mid-Pliocene simulations from CCSM4 to CESM2. https:/doi.org/10.1002/essoar.10501546.1

Done. Thank you.

17. Line 304 to 305: It might worth noting that this is different from the recent PlioMIP2 runs with CCSM4/CESM1 and 2 (the study listed in the above) (heat transport are reduced in both hemispheres).

This, along with a reference to another model, has been added to the text.

18. Line 350 – 351, "It is worth noting that not only does Eoi280 give the best fit at low latitudes, but bucks the trend at the northern high latitudes where the discrepancy is smaller than that for Eoi350." I don't understand this sentence.

We have changed the sentence to give a better explanation. At northern high latitudes, as $CO_2$ is decreased from 450ppm to 350ppm, the magnitude of the discrepancy between model SST anomaly and PRISM3 SST anomaly increases from 1.40°C to 4.43°C. However, a further decrease in $CO_2$ to 280ppm leads to a reversal and a smaller discrepancy of 3.03°C.

19. Line 390-391, "with modelled climate, ice sheets and vegetation exhibiting strong regional variations associated with orbital parameters, whether as time-dependent forcing in transient simulations (Willeit et al., 2013) or fixed to minimum or maximum forcings (Dolan et al., 2011)." Feng et al., (2017, EPSL) further demonstrated the regional warming/cooling patterns induced by mPWP orbital variations in a statistically meaningfully way.

This paper has now been cited in the text.

---

## Author Comment (AC2) · 9 May 2020

We thank the reviewer for his/her comments which we found most useful and have added much to the discussion. Below, we reply to each of the comments which are in red font.

General comments

1. The authors have included the EPlio1 experiment in the present study to compare with the PlioMIP2 core experiment. However, it lacks broadly discussion about the difference between these two experiments with regards to the SAT, SST, precipitation and sea ice. The Eplio1 differs from the PlioMIP2 not only in the topography, which is mostly mentioned in the results, but also in the land-sea mask, the lakes in the Africa as well as the soil types. I suggest the authors could pay more attention with regards to these aspects.

We have expanded our discussion on the differences between $Eoi^{400}$ and Eplio1 at the end of each subsection and included a supplementary figure to show the distribution of the differences. Any differences in the climate resulting from changes in soil type or the inclusion of Pliocene lakes in Africa appear to be very subtle, and an analysis would probably be better served with sensitivity experiments targeting these conditions individually.

2. Section 4.6: The Meridional heat transport have been studied in many PlioMIP2 papers (Li et al.,2020, Chandan and Peltier et al.,2017, Tan et al.,2020, Feng et al.,2020 etc). I suggest the authors can compare the results with the others. According to Figure 13, it is worth noting that the import change appears over the low latitudes which is different from the aforementioned studies. By the way, Figure 13 is not easy to read, I suggest to put all the absolute values in one plot.

As suggested, we have compared the meridional heat transport in our model to those with other PlioMIP2 models. This has been done for both the Atlantic Ocean and the total globe (atmosphere plus ocean), wherever possible, although in some studies, the anomalies are not shown explicitly. We have also made a note of the fact that the largest anomalies in the total meridional heat transport occur at the low latitudes, something that is not seen in other studies. We have put all the absolute values in both Figures 12(a) and 13(a). However, because the magnitudes of the differences for the total transport are small compared to the absolute values, it is not very easy to distinguish between the various experiments in Figure 13(a), particularly the non-core experiments.

3. Section 4.7: I do not understand why the authors compare the E280 with the proxy data reconstructed for the mid-Pliocene and discuss the performance of the data-model fit of the Pliocene experiment and the Pliocene experiment together (line 316). Section 4.8: I suggest the authors could also compare the model outputs with the PRISM4 data (Foley and Dowsett, 2019) which is more specific for the PlioMIP phase 2.

Both $E^{280}$ and $Eoi^{400}$ (in addition to the other Pliocene experiments) need to be shown in the comparison because the proxy data refer to absolute values of the temperature, and not the mid-Pliocene – Pre-Industrial anomalies. Without plotting the $E^{280}$ values, it is not possible to know whether the inclusion of Pliocene boundary conditions leads to better agreement with the proxy data. In our original manuscript, we compared results from

$Eoi^{400}$ with PRISM4 data and the results were plotted in Supplementary Figure 2 which we will now move to the main part of the manuscript. We have now expanded the text to include a comparison with the Pliocene experiments with other $CO_2$ levels, except Eplio1. We find that, for all $CO_2$ levels used in the study, not only is there still an underestimation of the warming in the northern North Atlantic Ocean and Greenland and Norwegian Seas, but also in the South Atlantic Ocean, near southern Africa.

Specific comments

1. Line 145: Confused. Please specify which 4 experiments are further integrated for another 1000 years and why they need to be integrated for another 1000 years

That sentence (and the whole of section 3.2) refers to the experiments, $Eoi^{280}$, $Eoi^{350}$, $Eoi^{400}$ and $Eoi^{450}$. They need to be integrated for another 1000 years because, up to that point, the values of $CO_2$ and the other greenhouse gases are slightly different to those specified in the PlioMIP2 protocols (Haywood et al, 2016). The values then follow the protocol for a 1000-year model integration, as plotted on the right-hand side of figure 2. This issue is related to the point directly below.

2. Line 155: Why are the greenhouse gases not changed to be PlioMIP level in the first 3000 years run?

Our original intention was to remain consistent with our previous study for the first phase of PlioMIP which did not explicitly state the Pre-Industrial greenhouse gas levels. This was left to each modelling group. For example, our Pre-Industrial $CO_2$ was set to approximately 285ppm, and double $CO_2$ was based on this value. All 8 experiments based on these greenhouse gas levels are shown on the left-hand side of figure 2. However, it soon became clear that all other modelling groups were using levels specified in PlioMIP2. In order to be consistent with them and facilitate model intercomparison, we decided to continue these experiments with the exact PlioMIP2 greenhouse gas levels for another 1000 years. This was preferable to restarting from scratch because the first set of experiments (on the left-hand side of figure 2) had already reached a state of near-equilibrium.

3. Line 167: In my opinion, figure 3a does not approve "Temperature increase fairly uniformly", In figure 3a, the warming amplitude over high latitudes is stronger than the low latitudes, the land warming is larger than the ocean broadly, and some extreme warming regions are found in the Barents sea and around coastal regions of the Antarctic.

Yes, we agree. In comparison to the Pliocene experiments, temperature changes in $E^{400}$ are not as extreme, but, in hindsight, we should have avoided the word 'uniform'. The text has now been changed. A description of the greater warming over land, in the Barents Sea and in some coastal parts of Antarctica was already given at the beginning of section 4.1.

4. Line 184: Compared to the PlioMIP1, PlioMIP2 also modify the Bering strait and Northern Canadian Archipelago regions, do they have any impacts on this different warming?

Cooler water in the Arctic Ocean is prevented from flowing into the Labrador Sea via the Northern Canadian Archipelago regions in PlioMIP2. This leads to warmer SST in the Labrador Sea. However, surface air temperature over the Labrador Sea remain the same or actually increases slightly because of the influence from the higher elevation over southern Greenland and North America in PlioMIP2. This higher elevation has a large influence on the northern hemisphere surface air temperatures. To better quantify the impact of closing the straits on the difference in warming would require a separate PlioMIP2 experiment with open straits.

5. Line 196: "near-uniform increase outside the polar regions" is not appropriate, according to Figure 4, I suggest to be "near-uniform increase outside the low latitudes (30N-30S)".

We have changed this part to 'a small but gradual and near-linear increase in temperature anomaly starting from the southern mid-latitudes to the northern mid-latitudes'.

6. Line 194: What is the "bias" here?

We have removed the word 'bias' and referred to the larger temperature anomaly north of the equator compared to south of the equator.

7. Line 205: What is the reason for the seasonal change of the arctic region, is the sea ice fraction change responsible for that?

Yes, the seasonal changes in the SAT of the Arctic region are related to the changes in the sea ice. In the summer, there is very little to no sea ice in the Arctic in $Eoi^{400}$ and so the ocean warms up more from incoming insolation. The SST anomaly is at its maximum during the summer. As the summer ends, heat from the ocean is released into the atmosphere. Since there is practically no summer sea ice in $Eoi^{400}$, more heat can be released, explaining the greater surface air temperature anomaly in the Arctic during September to November. This was also seen in the study by Zheng et al (2019):
Zheng, J., Zhang, Q., Li, Q., Zhang, Q. and Cai, M.: Contribution of sea ice albedo and insulation effects to Arctic amplification in the EC-Earth Pliocene simulation, Clim. Past, 15, 291–305, https://doi.org/10.5194/cp-15-291-2019, 2019.

8. Line 221: What is the potential reason for the SST anomalies for the other regions, e.g. the Labrador sea, the North Atlantic Ocean?

The SST in the Labrador Sea is warmer in $Eoi^{400}$ than in Eplio1 because the latter case had open Canadian Arctic Archipelago Straits (CAAS), and cooler water flowed from the Arctic Ocean to the Labrador Sea via the CAAS. Warmer SST in most parts of the North Atlantic and cooler SST further north, near Iceland are also seen in previous sensitivity experiments (not part of the present study) whereby we closed the CAAS and Bering

Strait in a Pre-Industrial scenario. Changes in ocean circulation led to these changes in SST, similar to those seen in similar sensitivity experiments performed in Otto-Bliesner et al. (2017).

---

## Author Response (AR1)

Dear Aisling,

In response to your question, I used the surface air temperature reconstructions at marine sites from Salzmann et al. (2013) only, as I did not want to make the manuscript too long; furthermore one other modelling group used the same data and this provided an opportunity to see how both models compared with the same proxy data sets. However, having given some more thought to your question, I think it would be more appropriate to include a comparison with the other data set which refers to terrestrial vegetation data. In addition to the revisions made in response to the reviewers' comments, I have now included a new figure for comparisons with this terrestrial vegetation data set and an extra paragraph at the end of section 4.7 to discuss the contents of this figure.

My responses to the reviewers have previously been posted on the site. Below are all the changes made to the manuscript.

In the supplement, figures 1 and 3 are new. Figure 2 is the same as figure 1 in the version first submitted.

Thank you,

Wing-Le

[revised manuscript text omitted]